# CONTROLLING INFORMATION LEAKAGE IN CONCEPT BOTTLENECK MODELS WITH TREES

## ABSTRACT

As AI models grow larger, the demand for accountability and interpretability has become increasingly critical for understanding their decision-making processes. Concept Bottleneck Models (CBMs) have gained attention for enhancing interpretability by mapping inputs to intermediate concepts before making final predictions. However, CBMs often suffer from information leakage, where additional input data, not captured by the concepts, is used to improve task performance, complicating the interpretation of downstream predictions. In this paper, we introduce a novel approach for training both joint and sequential CBMs that allows us to identify and control leakage using decision trees. Our method quantifies leakage by comparing the decision paths of hard CBMs with their soft, leaky counterparts. Specifically, we show that soft leaky CBMs extend the decision paths of hard CBMs, particularly in cases where concept information is incomplete. Using this insight, we develop a technique to better inspect and manage leakage, isolating the subsets of data most affected by this. Through synthetic and real-world experiments, we demonstrate that controlling leakage in this way not only improves task accuracy but also yields more informative and transparent explanations.

## 1 INTRODUCTION

Deep learning models have demonstrated significant capabilities and been widely adopted in applications such as image recognition, natural language processing, and disease prediction. However, their use in high-stakes fields like healthcare, pollution monitoring, credit risk, and criminal justice has raised concerns about transparency and flawed predictions (Hu et al., 2019). While numerous methods provide post-hoc analysis of trained neural networks (e.g., Ghorbani et al. (2019); Zhou et al. (2018)), these explanations do not always align with human understanding (Rudin, 2019).

Several recent studies suggest explicitly aligning intermediate outputs of neural network models with predefined expert concepts during supervised training processes (e.g Koh et al. (2020); Chen et al. (2020); Kumar et al. (2009); Lampert et al. (2009)) through the use of Concept Bottleneck Models (CBMs). Given a high-dimensional input of features (such as the raw pixels of an image), CBMs first predict a set of human-understandable concepts, which are then used to predict the final task labels with the help of an interpretable label predictor. Thus, a user is able to follow the decision-making process of the label predictor, while the task performance can remain close to that of the black-box model (Koh et al., 2020).

Despite these characteristics, CBMs frequently suffer from *information leakage* (Mahinpei et al., 2021; Margeloiu et al., 2021). This occurs where unintentional signal from the data not captured by the concepts is used for predicting the label and potentially increases its accuracy. If the label predictions rely on this information, explanations derived from the concepts may be inaccurate and potentially misleading. Crucially, leakage compromises our ability to intervene on concepts – a key benefit of CBMs. Recent works have attempted to mitigate leakage in CBMs by allowing a residual layer or side channel to capture a set of unknown, *latent concepts* (e.g. Havasi et al. (2022a); Shang et al. (2024); Zabounidis et al. (2023); Heidemann et al. (2023); Vandenhirtz et al. (2024)). Yet, while leakage is reduced, the information captured by these latent concepts is difficult to interpret and not necessarily disentangled from the known concepts, only partially resolving the issue.

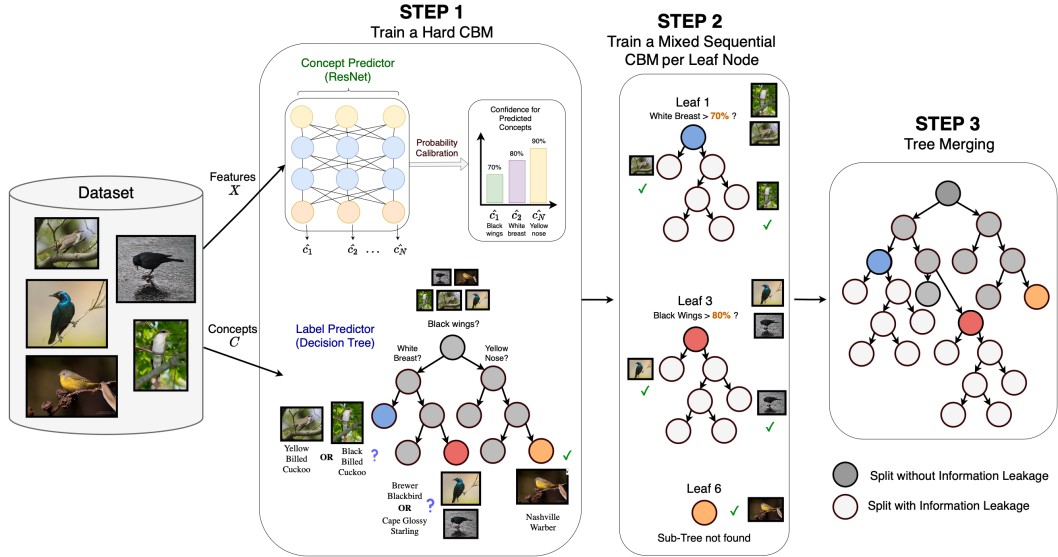

Figure 1: An overview of the Mixed CBM Algorithm (MCBM). *Step 1*: Two independent networks, a concept predictor with calibrated probability outputs and a decision tree label predictor (global tree) are trained. *Step 2*: A Sequential CBM (sub-tree) with mixed concept representations (*Mixed CBM*) further splits the leaf nodes of the global tree that present missing concept information and are prone to Leakage. *Step 3*: All trees are merged for global Leakage Inspection.

Given the current limitations, the purpose of this work is provide an interpretable method for inspecting where leakage occurs in CBMs and controlling this leakage using decision trees. We develop a method called Mixed CBM Training with Trees (MCBM) for better inspecting and managing leakage in those subsets, and introduce both sequential (MCBM-Seq) and joint training (MCBM-Joint) variants of this method for different scenarios of incomplete concept sets. An overview of our architecture can be found in Figure 1. Due to their hierarchical nature, trees allow us to first identify leaf nodes with subsets that are missing concept information, and subsequently control information leakage to only specialise those decision paths. This is achieved through a 3-step process, where a Hard CBM (global tree) is first trained, then an individual sub-tree is trained to extend each leaf node only if it can take advantage of leakage, and finally all trees are merged for global inspection.

**Our contributions are as follows**: We introduce a tree-based method for inspecting and controlling leakage in CBMs. We show that our method enables more interpretable decision-making with explanations that have higher accuracy and are guaranteed to make faithful predictions (in terms of fidelity) when concept sets are incomplete. We also demonstrate that our method allows us to quantify leakage for specific data subsets associated with tree regions where concept information may be incomplete, while identifying those decision rules most affected by leakage. Finally, we show that the derived group-based explanations can be very meaningful in real-life decision-making scenarios, and provide practical recommendations for selecting the appropriate method for training a CBM, based on the context of a problem and preferences of the user.

## 2 RELATED WORK

**Concept Bottleneck Models and Information Leakage.** Concept Bottleneck Models (CBMs) (Koh et al., 2020; Lampert et al., 2009; Kumar et al., 2009) are trained on data with covariates $x \in X$, target $y \in Y$, and annotated binary concepts $c \in C$. These models use a neural network $f_\theta$, parameterized by $\theta$ and structured as $\langle g_\psi, h_\phi \rangle$ (Leino et al., 2018), to enforce a concept bottleneck $\hat{c} = h_\phi(x)$. The final output depends solely on the predicted concepts $\hat{c}$. Soft CBMs (Chen et al., 2020; Koh et al., 2020) improve prediction by using probabilistic concept values but are prone to information leakage from the concept predictor to the label predictor (Margeloiu et al., 2021; Mahinpei et al., 2021). Most CBM research focuses on extending concept representations in the

embedding space to enhance predictive power (Zarlenga et al., 2024; Oikarinen et al., 2023; Kim et al., 2023; Semenov et al., 2024), while neglecting information leakage. Some works address leakage by allowing missing concept information to bypass concept representations through a Residual Layer (Yuksekgonul et al., 2023; Shang et al., 2024). Havasi et al. (2022b) tackle missing information with a side channel and an auto-regressive concept predictor, but these approaches struggle with interpretability and disentanglement of residual information (Zabounidis et al., 2023). Another approach by Marconato et al. (2022) rejects test samples prone to leakage, though it relies on assumptions about the prior distribution. Unlike these approaches, the work we present here enables both interpretability and leakage inspection, without any prohibitive assumptions on the distribution that may not hold in practice.

**Explainable Label Predictors.** According to CBM definitions (Koh et al., 2020), any interpretable machine learning model can serve as a label predictor, such as Logistic Regression (McKelvey & Zavoina, 1975), Generalised Additive Models (Hastie & Tibshirani, 1985), or Decision Trees (Breiman et al., 1984; Kass, 1980). Recent work integrates neural networks with interpretable decision-making. Wu et al. (2018; 2020) introduce tree-regularization to approximate neural network boundaries with decision trees. Ciravegna et al. (2021) propose the $\psi$ network for logic-based concept explanations using L1-regularization and pruning to extract interpretable First Order Logic formulas. Barbiero et al. (2022) enhance this with the Entropy-Net, using Entropy Loss for more concise logic formulas. Ghosh et al. (2023) further introduce a mixture of Entropy-Net experts for specialized explanations while maintaining performance. Yet, these concept-based explanations assume concept probabilities without evaluating potential label information leakage as we propose.

## 3 PRELIMINARIES

**Problem Setting.** We consider a classification task with $N = \{1, ..., n\}$ samples, $K = \{1, ..., k\}$ concepts and $R = \{1, ..., r\}$ classes. We assume a training set $D_{train} = \left\{ \left( x^{(i)}, c^{(i)}, y^{(i)} \right) \right\}_{i=1}^{N}$, where: $x^{(i)}$ is an input feature vector (e.g. an image) of the input space $X \subset \mathbb{R}^d$; $c^{(i)}$ is a categorical vector of $k$ concepts of the concept space $C \subset \{0, 1\}^k$; $y^{(i)}$ is a one-hot encoded vector of the target space $Y \subset \{0, 1\}^r$. A test set $N_{test}$ with $X_{test} \subset \mathbb{R}^d$ is also given, without annotated concepts.

**Concept Bottleneck Models.** The architecture of a CBM first introduces a *concept predictor* $f(W_1) : X \rightarrow C$ that maps inputs to concepts. It then uses a *label predictor* $g(W_2) : C \rightarrow Y$ that maps concepts to targets. This is typically any interpretable model, such that the relationship from concepts to targets can be explained e.g linear layer decision trees. CBMs can be classified into two categories (Havasi et al., 2022a; Koh et al., 2020):

*Hard CBMs*: At test time, the label predictor only accepts binary (hard) concepts as inputs. The networks $f$ and $g$ are trained independently on the ground truth data.

$$L_C = L_{W_1}(\hat{C}, C) = \sum_i L_{W_1}\left( f(x^{(i)}) ; c^{(i)} \right) = \sum_{i,k} L_{W_1}\left( f(x^{(i)})[k] ; c^{(i)}[k] \right), k \in K \quad (1)$$

$$L_Y = L_{W_2}(\hat{Y}, Y) = \sum_i L_{W_2}\left( g(c^{(i)}) ; y^{(i)} \right) \quad (2)$$

We use cross-entropy loss as the task loss $L_Y$. For the concept loss $L_C$, we use the sum of binary cross-entropy losses for independent concepts, with the sum of cross-entropy losses for groups of mutually-exclusive concepts. At test time, we make a prediction for a sample $x_*$ by first converting the predicted logits into concept probabilities $\hat{c} = \sigma(f(x_*))$, where $\sigma$ is either a sigmoid function for independent concepts or a softmax function for mutually-exclusive concepts. We convert the probabilities to binary representations $\hat{c}_{bin}$ either through thresholding (sigmoid) or the *argmax* operator (softmax), and we pass them to the label predictor: $y_* = g(\hat{c}_{bin})$.

*Soft CBMs*. At test time, the label predictor accepts concept probabilities (soft concepts) as inputs. These can be trained either independently, sequentially or jointly. Independent training uses a procedure identical to Hard CBMs, based on Eq. 1, 2. At test time, we use the concept probabilities: $y_* = g(\hat{c})$, where $\hat{c} = \sigma(f(x_*))$. For sequential training, the network $f$ is trained according to the objective of Eq. 1, and then the predicted concepts $\hat{c}^{(i)} = \hat{g}(x^{(i)})$ are used as input to train the network $g$, minimising the loss: $L_Y = L_{W_2}(\hat{Y}, Y) = \sum_i L_{W_2}\left( g(\hat{c}^{(i)}) ; y^{(i)} \right),$ where $\hat{c}^{(i)} =$

$\sigma(f(x^{(i)}))$. Joint training trains both networks simultaneously. The hyper-parameter $\lambda_C$ controls the relative importance of the two tasks. Assuming $\hat{c}^{(i)} = \sigma(f(x^{(i)}))$:

$$L = L_Y + \lambda_C L_C = \sum_i L_{W_1,W_2}\left(g(\hat{c}^{(i)})\,;\,y^{(i)}\right) + \lambda_C \sum_i L_{W_1}\left(f(x^{(i)})\,;\,c^{(i)}\right) \qquad (3)$$

**Leakage in CBMs.** While information leakage is defined by Mahinpei et al. (2021) as a type of unintended information that the concept representations capture, in this work we propose a more explicit definition:

**Definition 3.1** *The amount of unintended information that is used to predict label $y$ with soft concepts $\hat{c}$ that is not present in hard representation $c$.*

In our work, we measure this as the *mutual information* of targets $y$ and soft concepts $\hat{c}$ given the hard concepts $c$,

$$I_{\text{Leakage}} \;=\; I(y;\hat{c}\,|c) = H(y|c) - H(y|\hat{c},c) \qquad (4)$$

## 4 Tree-Based Leakage Inspection and Control

In what follows, we present the core contribution of our work. Specifically, given both a trained hard CBM and a trained (either sequentially or jointly) soft CBM, we would like to determine whether it is possible to inspect any information leakage that the soft model exploits to make its predictions, and understand how this information was used. Based on Definition 3.1, we know that any additional information used to predict $y$ that is not contained in concepts $c$, should be captured by the difference of conditional mutual information terms in Eq. 4. A naive attempt to answer this question is thus to first train the same concept predictor for both hard and soft models, and then use separate decision tree classifiers for each CBM as label predictors and subsequently inspect their trees to observe how they differ. However, inspecting the corresponding decision tree label predictors for both hard and soft CBMs is non-trivial as the trees be incomparable and contain very different splits to distinguish samples (an example is given in Appendix A.3).

Motivated by this challenge, we present a new training method with *mixed concept representations*. Our key insight is that any subset of data that cannot be further split by a hard CBM's decision tree but can be split by a soft leaky CBM's tree is vulnerable to information leakage. The method has three steps, shown in Fig. 1. First, we train a *hard, leakage-free* CBM with a decision tree as the *global* label predictor. This tree is decomposed into decision paths with some concepts, while other concepts are predicted independently with calibrated probabilities. In the second step, the decision paths are used to train a *mixed CBM*, combining hard and soft concepts with a new tree-based label predictor. Finally, the global tree and sub-trees are merged for complete inspection.

### 4.1 The Mixed Sequential CBM (MCBM-Seq) Training Algorithm.

**Identifying samples corresponding to certain concepts.** We train the concept predictor $f : X \to C$ according to Eq. 1 and the label predictor $g : C \to Y$ independently. In principle, the concept predictor could be any deep learning model; we replace the label predictor of a standard CBM by a decision tree (Breiman et al., 1984; Kass, 1980). Specifically, the tree is constrained to a minimum number of samples per leaf ($msl$) to prevent overfitting. We refer to this tree as the "global" tree containing decision paths with all the concepts. We then decompose the tree into a set of decision paths $T_{hard} = \{P_1, ..., P_M\}$, where $M$ is the number of leaf nodes. Each decision path corresponds to a set of binary decision rules that are not affected by leakage, since the tree was trained on the ground truth binary concepts. Fig. 2 depicts these rules intuitively as a set of True or False questions used to distinguish a test sample from other classes.

**Calibrating the concept probabilities to prevent overconfident predictions.** To overcome the problem of overconfidence in the predictions of deep neural networks (Guo et al., 2017), we calibrate the predicted concept probabilities of the trained concept predictor. We perform Platt scaling (PLATT, 1999) for binary (independent) concept predictions, and temperature scaling for multi-class (mutually-exclusive) concept predictions (Guo et al., 2017). We argue that the calibration step is crucial for the interpretability of Sequential CBMs (which are trained in the next step), since the

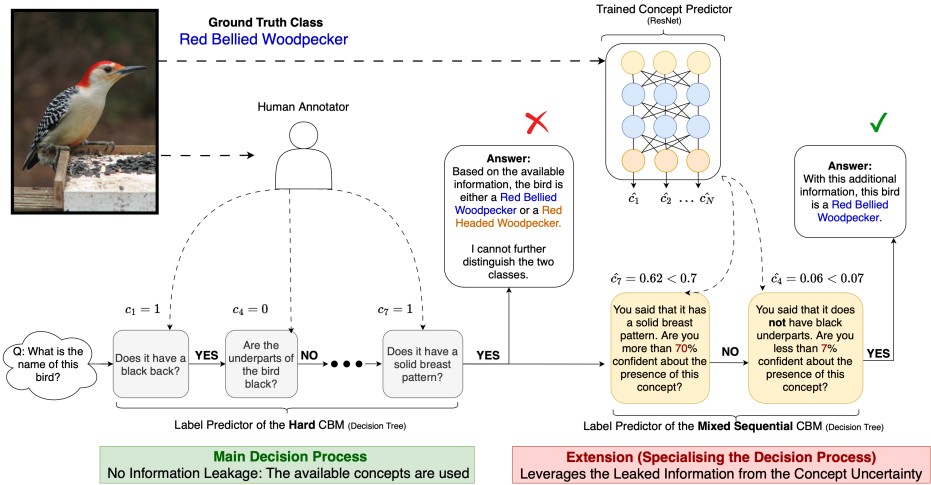

Figure 2: Summary of the Decision Making Process of The Mixed Sequential CBM Algorithm (MCBM-Seq) when classifying an image with annotated concepts. The process is described intuitively as a conversation between the Tree label predictor and two entities that provide the input concepts: the Human Annotator and the Concept Predictor. The concept probabilities are used to specialise the decision process only when the available annotated concepts are not sufficient.

decision rules of the label predictor are based on the true concept probabilities and can often be human intuitive (see section A.10).

**Training a mixed sequential CBM for each decision path.** We first isolate the set of all concepts $K_m \subseteq K$ used in the decision splits of each path $P_m$, and the set of the remaining concepts $K'_m = K - K_m$ not used in the path. Then, for each decision path in the global tree, we train a mixed sequential CBM. Specifically, we extract a subset of the dataset $\{X_m, C_m, Y_m\}$ with the training samples classified by the decision path $P_m$. We use the already trained network $f : X \rightarrow C$ as the concept predictor of the CBM. Then for each sample $i$, we construct a new concept vector $c_i^*$ as follows: For the concepts appearing in the decision path, we assign the calibrated soft probability given by the concept predictor: $c_i^* [k] = \hat{c}_i [k] = f(x_i)[k], \forall k \in K_m$. For the remaining concepts, we assign the hard (binary) value: $c_i^* [k] = c_i [k], \forall k \in K'_m$. Since each concept vector does not contain exclusively hard (binary) concept values or soft concept probabilities, but a combination of both, we name this architecture a *Mixed CBM*. Next, we train a new Decision Tree as the label predictor of the CBM, however constrained on the *same number of minimum samples per leaf as the global tree (msl)*. This ensures that the new sub-tree, if found, can further specialise the data *only* because of the additional leaky information encoded in the concept probabilities, and not because of a smaller constraint in the number of samples allowed per leaf.

We should emphasise that the concept predictor is only trained *once*, and thus our method does not introduce any computational overhead compared to a Sequential CBM with a single decision tree as label predictor (see section A.4). A second important detail is that we train a CBM with *mixed* concept representations per leaf subset, and not a purely Soft Sequential CBM. This allows us to investigate if the soft representation of one or more of the concepts *that have appeared in the decision path of this subset in the global tree* could further specialise the decision process. This is crucial in order to quantify leakage in section 4.2 and to approximate our definition of leakage in Eq. 4, because only the concepts $K_m$ present in the non-leaky decision path of a leaf $m$ in the global tree are those shared by all samples $s$ in the leaf, and thus satisfy the definition of the conditional entropy for this leaf: $H(y_s|c_k), \forall k \in K_m$. Instead, if we trained a purely Soft CBM, some of the concepts may not be shared by this group. Referring again to the example of Fig. 2, we would like to observe how a decision rule based on the confidence of the concept predictor (shown in a yellow box) can specialise one or more binary rules that appeared in the global tree (shown in a white box).

**Merging the sub-trees to the global tree**. This final step is optional and only used when the size of the tree is reasonable and the user would like to grasp a global picture of label classifications

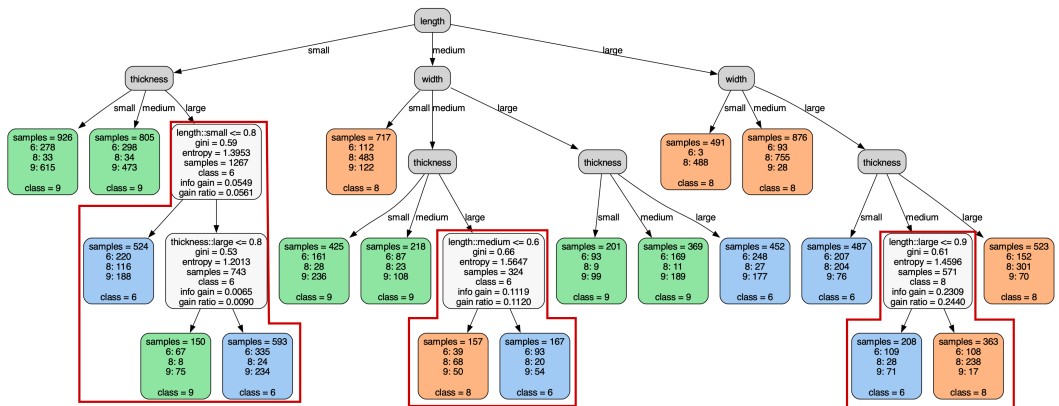

Figure 3: The MCBM-Seq algorithm for a reduced Morpho-MNIST dataset with digits 6, 8 and 9 and concepts "length", "thickness" and "width". The final tree merged the sub-trees is shown. If a sub-tree is found, it replaces the leaf node of the hard CBM and is highlighted in a red box. The remaining leaf nodes are unaffected by leakage. This architecture allows us to both inspect and restrict leakage only to subsets with missing concept information.

and Leakage. For group-specific explanations, an analysis per individual decision path can be more informative (see section 5.3). A visualisation of a merged tree after performing the algorithm is provided in Fig. 3. At *test time*, we retrieve the decision path that classifies each sample in the *global* tree, and make a prediction using the corresponding soft tree. The complete pseudo-code is provided in Algorithm 1.

## 4.2 QUANTIFYING INFORMATION LEAKAGE WITH TREES

The MCBM-Seq algorithm allows us to inspect and quantify Information Leakage. Consider a subset of samples $s$ that end up in one of the leaf nodes of the global tree, and consider a soft concept $\hat{c}_k$ based on which we perform the first split in the Mixed Sequential CBM. Thus, the subset $s$ is divided into two new subsets, $s_1$ and $s_2$, and the new tree has three nodes in total. We also assume the target distribution of subsets $s$, $s_1$ and $s_2$ are $y_s$, $y_{s_1}$ and $y_{s_2}$ respectively. We can use the Information Gain (Breiman et al., 1984) we achieve when splitting a node with a soft concept $\hat{c}_k$ as a measure of the Information Leakage that the soft concept provides, based on Eq. 4:

$$ I_{\text{Leakage}}(\hat{c}_k) \approx H(y_s) - \left[ \frac{|s_1|}{|s|} H(y_{s_1}) + \frac{|s_2|}{|s|} H(y_{s_2}) \right] = IG(\hat{c}_k) \tag{5} $$

**This is only valid because of the way we constructed our tree**, by first using the hard concepts to fit the global tree and then specialising the leaf nodes with soft concept splits. Also, this formulation allows us to inspect and quantify leakage **specifically for each split**, by adjusting the concept values $c_s$ and $\hat{c}_k$ accordingly. Refer to Appendix A.2 for more details.

## 4.3 MIXED JOINT CBM TRAINING WITH TREES (MCBM-JOINT)

Joint CBMs tend to achieve higher task performance compared to the other forms of CBM Training, due to the end-to-end optimization procedure of Eq. 3 (Koh et al., 2020). However, this comes at the expense of interpretability. According to (Mahinpei et al., 2021), they are more prone to Information Leakage compared to Sequential CBMs because the label predictor can "shape" the concept probabilities in an unexplainable way in order to improve the task performance while decreasing concept accuracy. In practice, this trade-off is controlled by the $\lambda_C$ parameter of Eq. 3. Unlike Sequential CBMs, these probabilities do not correspond to the true confidence of the concept predictor, and thus concept-based explanations based on these probabilities are even less human-intuitive. However, we could control Information Leakage to only specialise the main decision rules, in order to achieve better interpretations when higher performance is a priority.

---

**Algorithm 1** Mixed Sequential CBM Training (MCBM-Seq)

---

**Input:** $N$ samples; $K$ concepts; $R$ classes; A dataset $D_{train} = \{X, C, Y\}$, where $X \subset \mathbb{R}^d$, $C \subset \{0,1\}^k$,
$\quad\quad Y \subset \{0,1\}^r$; A set of $N_{test}$ samples with $X_{test} \subset \mathbb{R}^d$; Minimum Samples per leaf $(msl)$.

**Output:** A hard tree: $T_{hard}$; a set of soft trees, one for each leaf: $T_{soft} = \{T_1, ..., T_M\}$; test predictions $\hat{Y}$.

1: **procedure** TRAINING$(D_{train}, msl)$
2: $\quad$ Train the concept predictor $f : X \rightarrow \hat{C}$, where $\hat{C} \subset [0,1]^k$ (Eq. 1);
3: $\quad$ Define a decision tree constrained on the min samples per leaf $T_{hard} = Tree\,(msl)$;
4: $\quad$ Train the tree on the hard concept set: $T_{hard}.fit(C, Y)$;
5: $\quad$ Decompose the tree into a set of $M$ decision paths $T_{hard} = \{P_1, ..., P_M\}$;
6: $\quad$ Train a sub-Tree per leaf, by calling the procedure: $T_{soft}$ = TRAINING SUB-TREE$(D_{train}, f, T_{hard})$;
7: $\quad$ **return** $T_{hard}, T_{soft}$
8: **end procedure**
9:
10: **procedure** TRAINING SUB-TREE$(D_{train}, f, T_{hard})$
11: $\quad$ **for** each decision path $m \in M$ **do**
12: $\quad\quad$ Collect the data for the samples of $P_m$: $D_m = \{X_m, C_m, Y_m\}$, $X_m \subseteq X$, $C_m \subseteq C$, $Y_m \subseteq Y$;
13: $\quad\quad$ Get the concept probabilities for the samples of the path $\hat{C}_m = f(X_m)$;
14: $\quad\quad$ Calibrate the concept probabilities using Platt or Temperature scaling
15: $\quad\quad$ Isolate the set of concepts $K_m \subseteq K$ used as splits in the path $P_m$;
16: $\quad\quad$ Isolate the set of concepts $K'_m = K - K_m$ not used as splits in the path $P_m$;
17: $\quad\quad$ **for** each sample $i \in N_m$ **do**
18: $\quad\quad\quad$ Initialise a new concept vector for the sample $c_i^*$;
19: $\quad\quad\quad$ Assign the soft concept values of $i$ for the concepts used in the path: $c_i^*[k] = \hat{c}_i[k]$, $\forall k \in K_m$
20: $\quad\quad\quad$ Assign the hard concept values of $i$ for the remaining concepts: $c_i^*[k] = c_i[k]$, $\forall k \in K'_m$
21: $\quad\quad$ **end for**
22: $\quad\quad$ Concatenate the concept vectors to create a new concept set for the path: $C_m^* = \{c_1^*, ..., c_{N_m}^*\}$;
23: $\quad\quad$ Define a decision tree using the same $msl$ constraint: $T_m = Tree\,(msl)$;
24: $\quad\quad$ Train the new tree: $T_m.fit\,(C_m^*, Y_m)$;
25: $\quad$ **end for**
26: $\quad$ **return** $T_{soft} = \{T_1, ..., T_M\}$
27: **end procedure**
28:
29: **procedure** EVALUATION$(X_{test}, f, T_{hard}, T_{soft})$
30: $\quad$ Predict the concept values $\hat{C} = f(X_{test})$;
31: $\quad$ Get the decision paths for the test predictions $\{P_1, ..., P_M\} = T_{hard}.predict(\hat{C})$
32: $\quad$ **for** each decision path $m \in M$ **do**
33: $\quad\quad$ Isolate the test samples of the path $N_m \subseteq N_{test}, \hat{C}_m \subseteq \hat{C}$;
34: $\quad\quad$ Predict from the associated tree $\hat{Y}_m = T_m.predict(\hat{C}_m), T_m \in T_{soft}$
35: $\quad$ **end for**
36: $\quad$ **return** $\hat{Y} = \{\hat{Y}_1, ..., \hat{Y}_M\}$
37: **end procedure**

---

For this purpose, we propose the MCBM-Joint algorithm as a post-hoc analysis tool for trained Joint CBMs. The algorithm is identical with that of MCBM-Seq in section 4.1, but the concept probabilities are now extracted from the optimised concept predictor of the Joint CBM instead of an independently trained concept predictor. In contrast to MCBM-Seq, we do not calibrate these concept probabilities, since they do not correspond to the true confidence of the predictor. A visual intuition is shown in Appendix 7.

## 5 EXPERIMENTS

We evaluate the MCBM-Seq and MCBM-Joint methods in challenging image classification and medical settings, demonstrating the versatility of our approach across different metrics overall, as well as per decision path.

**Baselines**. We first compare our MCBM-Seq and MCBM-Joint methods with the standard modes of CBM training (Hard, Independent, Sequential) described in Koh et al. (2020), and we use a Decision Tree as the label predictor. For the MCBM-Joint method, we first optimise the concept

encoder with the Joint training objective of Eq. 3 using a simple linear layer as the label encoder. We also train all CBM methods using the state-of-the-art Entropy-Net Barbiero et al. (2022) as the label-predictor, which achieves higher task accuracy compared to linear label predictors while also providing interpretable logic explanations.

**Quantitative Metrics**. In terms of performance, we evaluate our CBM methods using the a) *Task* and b) *Concept* Accuracy (Koh et al., 2020). To compare the interpretability and reliability of our methods compared to existing work, we use the following metrics defined by Barbiero et al. (2022): a) The *Explanation Accuracy* as the task performance of a method when using its extracted explanation formulas, and b) the *Fidelity of an Explanation* which measures how well each formula matches the model's predictions. Finally, we provide an estimate of Information Leakage as the Information Gain of leaky splits according Eq. 5, which is unique to our methods.

## 5.1 DATASETS AND MODELS

**Morpho-MNIST** (Castro et al., 2018): The dataset describes MNIST digits in terms of measurable shape attributes which we use as concepts. These are the thickness, area, length, width, height, and slant of digits. We categorise each real-valued concept in one of three equally spaced bins, indicating a "small", "medium", "large" or value, resulting in mutually exclusive concept groups such as "length::small", "length::medium" and "length::large". We use a LeNet model Lecun et al. (1998) as the concept predictor. Hyper-parameter details are given in Appendix A.7.

**The CUB Dataset** (Koh et al., 2020): The dataset consists of $n = 11,788$ pictures of 200 different bird species. There are 312 annotated binary concepts available. Similarly to Koh et al. (2020); Yeh et al. (2020), we only select those concepts that appear in at least $10\%$ of the dataset, and thus we form a reduced set of 112 binary concepts. The data is denoised by converting instance-specific concepts to class-specific concepts via majority voting. To simulate a scenario of an incomplete concept set, we choose 45 concepts and denote the rest as missing. We train a ResNet-18 model (He et al., 2016) as the independently trained concept predictor (see Appendix A.6 and A.7 for pre-processing and hyperparameter tuning details).

**MIMIC**: MIMIC-IV (Medical Information Mart for Intensive Care IV) (Johnson et al., 2023) is a large, freely accessible dataset consisting of de-identified electronic health records from over 70,000 critical care patients. It includes data such as demographics, vital signs, laboratory results, medications, and clinical notes. Our binary classification task is to identify recovering or dying patients after ICU admission. Since the dataset does not have annotated concepts, we calculate the six Sequential Organ Failure Assessment (SOFA) scores (Lambden et al., 2019) and categorise them into 3 levels of severity, for a total of 18 concepts. We use a 3-layer MLP as the concept predictor. Details about the concept selection and hyper-parameters are given in Appendix A.6 and A.7.

## 5.2 OVERALL PERFORMANCE ACROSS BASELINES

**Our tree-based label predictor produces more trustworthy explanations compared to existing solutions when the concept sets are incomplete**. In their work, Barbiero et al. (2022) show that the *Explanation Accuracy* of the Entropy-Net (measured as the average F1 score across all label classes) matches the *Task Accuracy* in most experiments with almost perfect Fidelity scores. However, we observe in this work that this is not guaranteed for datasets with missing concept information. In Table 1, we observe that the fidelity scores of Entropy-Net are lower that $80\%$, since the logic formulae are missing concept rules and can thus associate some samples to multiple classes. The Explanation Accuracy drops for the same reason. On the other hand, Decision Trees are inherently rule-based models, where no such fidelity issues arise. While their overall Task Accuracy is lower according to Table 1, they exhibit higher Explanation Accuracy with perfect Fidelity. Our Mixed CBM methods allow for leakage inspection and control, to further assist the reliability of explanations.

**Mixed CBMs achieve higher task accuracy than their respective hard CBMs and less leakage than their soft counterparts**. This can be observed from the results of Table 2 and is expected because the MCBM-Seq and MCBM-Joint models are specialised versions of the same hard/independent CBM, with leaky information concentrated in the subtrees. Soft CBMs generally achieve better performance because they use the concept probabilities from the very first split of the root (see Appendix. 6), thus the Decision Tree finds the optimal splits without restrictions in

Table 1: Explanation Metrics for different Sequential CBMs. Tree-based label predictors achieve higher Explanation Accuracy for sets with incomplete concept information, and do not pose fidelity considerations. Our MCBM-Seq also allows for Leakage Inspection, compared to a Decision Tree.

| Method | Morpho-MNIST | | | CUB | | | MIMIC-III | | | Leakage Inspection |
|---|---|---|---|---|---|---|---|---|---|---|
| | Task% | Explanation% | Fidelity% | Task% | Explanation% | Fidelity% | Task% | Explanation% | Fidelity% | |
| Seq. (Entropy Net) | 52.26 | 25.01 | 77.81 | 65.23 | 42.37 | 46.87 | 83.34 | 68.94 | 56.11 | ✗ |
| Seq. (Decision Tree) | 48.42 | 48.42 | 100 | 55.59 | 55.59 | 100 | 83.05 | 83.05 | 100 | ✗ |
| **MCBM-Seq** | 49.62 | **49.62** | **100** | 47.39 | **47.39** | **100** | 82.05 | **82.05** | **100** | ✓ |

Table 2: Task and Concept Accuracy across different datasets and CBM training methods. Our Mixed CBM methods are comparable with current approaches in overall performance.

| | Method | Morpho-MNIST | | CUB | | MIMIC-III | |
|---|---|---|---|---|---|---|---|
| | | Task% | Concept% | Task% | Concept% | Task% | Concept% |
| Decision Tree | Hard | 47.20 | 89.94 | 46.51 | 94.82 | 81.65 | 98.67 |
| | Independent | 47.20 | 89.94 | 46.51 | 94.82 | 81.65 | 98.67 |
| | **MCBM-Seq** | 49.62 | 89.94 | 47.39 | 94.82 | 82.05 | 98.67 |
| | Sequential | 48.42 | 89.94 | 55.59 | 94.82 | 83.05 | 98.67 |
| | **MCBM-Joint** ($\lambda_C = 0.1$) | 83.16 | 83.38 | 56.55 | 94.21 | 82.55 | 97.15 |
| | **MCBM-Joint** ($\lambda_C = 1$) | 67.32 | 87.92 | 56.42 | 94.76 | 82.05 | 97.72 |
| | **MCBM-Joint** ($\lambda_C = 100$) | 50.72 | 90.58 | 56.29 | 94.99 | 81.85 | 97.82 |
| Entropy Net | Hard | 46.31 | 89.94 | 56.07 | 94.82 | 82.65 | 98.67 |
| | Independent | 45.10 | 89.94 | 53.74 | 94.82 | 82.55 | 98.67 |
| | Sequential | 52.26 | 89.94 | 65.23 | 94.82 | 83.34 | 98.67 |
| | Joint ($\lambda_C = 0.1$) | 98.17 | 83.38 | 72.26 | 94.21 | 83.74 | 97.15 |
| | Joint ($\lambda_C = 1$) | 95.02 | 87.92 | 69.37 | 94.76 | 83.44 | 97.72 |
| | Joint ($\lambda_C = 100$) | 56.07 | 90.58 | 64.12 | 94.99 | 82.85 | 97.82 |
| | Black-Box | 99.99 | - | 84.35 | - | 92.74 | - |

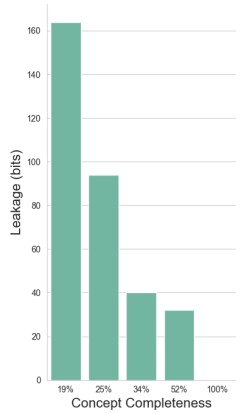

Figure 4: Leakage with Concept Completeness (CUB).

the architecture. However, this leads to Information Leakage in the whole tree and has undesirable effects for interpretability across all decision paths Unlike these, mixed CBMs constrain leakage only to specific leaf nodes, mitigating these undesirable effects throughout the rest of the tree.

**Information Leakage decreases when concepts completeness increases**. CBMs are more prone to Information Leakage as the number of missing concepts increases. While this issue was also raised by Havasi et al. (2022b), we are able to provide quantitative evidence by measuring the total Information Gain summed across all leaky splits of our Tree. Since the CUB dataset Wah et al. (2011) is concept-complete for class-specific explanations, we evaluate our MCBM-Seq method in Fig. 4 across different levels of completeness, by successively picking more concept groups from the dataset. In the absence of complete concepts, the global tree is small, with more subtrees containing leakage hence enabling more of the leaf nodes to be extended to improve the Information Gain.

## 5.3 PERFORMANCE PER DECISION PATH

**Our method enables inspecting Information Leakage per decision path**. A unique advantage of our method compared to existing CBM training methods is that we can extract performance metrics, inspect and control leakage per individual Decision Paths corresponding to particular groups of data. In Fig. 5, we decompose the full tree from the reduced Morpho-MNIST example in Fig.. 3 into fifteen decision paths, measuring the *Task Accuracy* against the original Hard CBM (global tree) in Table 3. For the three extended decision paths, the accuracy increased from $44\% \rightarrow 45\%$, $37\% \rightarrow 41\%$ and $44\% \rightarrow 57\%$ respectively due to leakage. We also report the Information Gain (Leakage) for each individual leaky split of a sub-tree, if found. We observe that the concept "length:large" in the leaky split of path 14 in Fig. 5 provides the largest Information Gain (0.230 bits) out of all leaky splits, suggesting that this concept could benefit from additional information. Repeating this for MCBM-Joint, we observe not all the same decision paths are necessarily extended with leakage. The method yields higher task accuracies but also higher information gain (in terms of leakage) Eg.In path 14, the respective leaky split of MCBM-Joint yields a very high accuracy of 80.65%, but the Information Gain (Leakage) from such split is also higher compared to MCBM-Seq (0.968 bits).

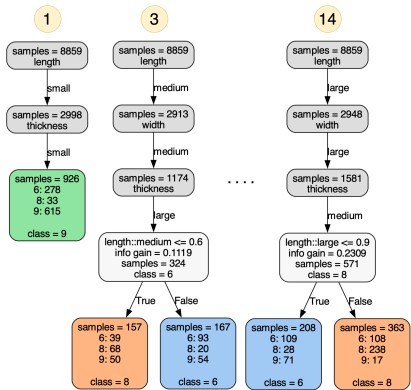

Figure 5: Decomposing the resulting trees of MCBM-Seq into Decision Paths for the Reduced Morpho-MNIST example of Fig. 3.

Table 3: Analysis per path on the Reduced Morpho-MNIST of Fig. 3. Acc. refers to Task Accuracy and IG refers to Information Leakage for each split of the "leaky" extension.

| Path | Hard | MCBM-Seq | | MCBM-Joint | |
|---|---|---|---|---|---|
| | Acc.% | Acc.% | IG | Acc.% | IG |
| 1 | 62.39 | 62.39 | - | 62.39 | - |
| 2 | 58.53 | 58.53 | - | 58.53 | - |
| 3 | **44.82** | **45.76** | [0.054, 0.006] | **60.45** | [0.191, 0.191] |
| 4 | 70.48 | 70.48 | - | 70.48 | - |
| 5 | 53.92 | 53.92 | - | 53.92 | - |
| 6 | 44.20 | 44.20 | - | 44.20 | - |
| 7 | **37.39** | **41.73** | [0.111] | 37.39 | - |
| 8 | 55.17 | 55.17 | - | 55.17 | - |
| 9 | 52.57 | 52.57 | - | **90.72** | [0.795] |
| 10 | 58.29 | 58.29 | - | **91.49** | [0.715] |
| 11 | 99.29 | 52.57 | - | 99.29 | - |
| 12 | 92.00 | 92.00 | - | 92.00 | - |
| 13 | 39.18 | 39.18 | - | **81.98** | [0.931] |
| 14 | **44.91** | **57.70** | [0.230] | **80.65** | [0.968] |
| 15 | 55.14 | 55.14 | - | **84.19** | [0.911] |

**Our tree-structure allows for meaningful group-specific explanations**. While instance-specific (Entropy-Net) Barbiero et al. (2022) and class-specific explanations can be too complex or generic, group-specific explanations are often more useful, especially in fields like healthcare. Our method allows intuitive control of group size via the minimum samples per leaf (msl) constraint (Appendix A.8). An example of a group-specific explanation using the MCBM-Seq method on the CUB dataset(Wah et al., 2011) is shown in Fig. 2. We consider two bird classes: Red Bellied Woodpecker and Red Headed Woodpecker. At test time, the method traverses the decision path in the global tree and identifies the two classes as indistinguishable based on available concepts. After training a Mixed-Sequential CBM, the label predictor separates the classes using the concept predictor's calibrated probability for "has-breast-pattern-solid." The predictor finds that Red Bellied Woodpeckers are less than 70% likely to have a solid breast pattern, while Red Headed Woodpeckers are more likely to possess this trait. The full decision path and case-study description are given in Appendix A.10. The user has three options: a) rely solely on the global tree for the most reliable prediction using majority voting, b) extend the decision process with MCBM-Seq for more intuitive and higher-performance results, or c) use MCBM-Joint's less intuitive probabilities for maximum accuracy. Importantly, unlike a purely soft CBM, leakage will not impact all decision-making paths in a mixed CBM and will be isolated to only some leaf nodes that can be extended.

## 6 CONCLUSION

In this work, we introduce MCBM-Seq and MCBM-Joint methods that use decision trees to inspect and control information leakage in CBMs. These tree-based approaches maintain high fidelity in the explanations and achieve better accuracy on datasets with incomplete concept information. Unlike purely Soft CBMs, the mixed concept representations limit information leakage in data subsets with insufficient concept information. They also quantify leakage per decision path and rule, producing more meaningful group-based explanations.

A limitation of our work is that the final, merged trees can be difficult to visualise and inspect for very large datasets. Yet, we show that a leakage analysis for specific decision paths of interest can also be meaningful. Also, *mixed* CBMs allow us to derive more interpretable and reliable explanations but typically do not match the model accuracy of purely Soft CBMs. A promising avenue for future work is to further address the problem of missing concept information by combining our approach with concept discovery strategies. While information leakage makes up for some of the missing concept information, these discovery strategies could be optimised per decision path in our tree-structure, to further distinguish groups of samples that cannot take advantage of leakage. Moreover, our method does not pose any constraint on the architecture of the concept encoder, thus it could also integrate a more expressive Auto-Regressive Concept encoder or a Stochastic Concept Encoder to further assist the concept predictions.

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

## A APPENDIX

### A.1 REPRODUCIBILITY STATEMENT

To ensure reproducibility, all experiments described in this paper were conducted with fixed random seeds, which are explicitly detailed in the provided code. In addition, all scripts necessary to replicate the main results will be made publicly accessible. Comprehensive instructions for running the experiments and verifying the outcomes will also be included in the repository, alongside the random seed values for each specific experiment. The anonymous code is given here: `https://anonymous.4open.science/r/ICLR_2025_Controlling_IL_With_Trees-75EA/README.md`

### A.2 USING TREES TO QUANTIFY INFORMATION LEAKAGE

As in section 4.2, consider again the subset of samples $s$ that end up in one of the leaf nodes of the global tree, and a soft concept $\hat{c}_k$ based on which we perform the first split in the Mixed Sequential CBM. Consider also the two subsets after the split $s_1$ and $s_2$, and the target distributions $y_s$, $y_{s_1}$ and $y_{s_2}$ respectively. According to Breiman et al. (1984), the **Information Gain** of a split in a decision tree can be defined as the difference of the entropy of the subset $s$ before the split and the weighted sum of the entropy's of the two nodes after the split using the concept $\hat{c}_k$:

If $c_s$ represents the set of hard concepts that appear in the decision path of the global tree up to the leaf node of the subset $s$, then we assume that

$$H(y|c_s) \approx H(y_s) \tag{6}$$

In other words, we have already taken into account the information present in the concepts $c_s$ to construct the target distribution of this path $y_s$ from the initial target distribution of all samples $y$ when building the global tree. Similarly, given we have the information from the hard concepts of the path $c_s$ and the new soft concept $\hat{c}_k$, we assume that:

$$H(y|\hat{c}_k, c_s) \approx \left[ \frac{|s_1|}{|s|} H(y_{s_1}) + \frac{|s_2|}{|s|} H(y_{s_2}) \right] \tag{7}$$

Thus, we can approximate the Information Leakage given to the label encoder by the soft concept $\hat{c}_k$ according to Eq. 4:

$$I_{\text{Leakage}}(\hat{c}_k) \; = \; I(y; \hat{c}_k \,|c_s) = H(y|c_s) - H(y|\hat{c}_k, c_s) \tag{8}$$

$$\overset{\text{equation 6 equation 7}}{\approx} \; H(y_s) - \left[ \frac{|s_1|}{|s|} H(y_{s_1}) + \frac{|s_2|}{|s|} H(y_{s_2}) \right] \tag{9}$$

$$\overset{\text{equation 5}}{=} \; IG(\hat{c}_k) \tag{10}$$

Thus, using Decision Trees, we can use the Information Gain we receive when splitting a node with a soft concept $\hat{c}_k$ as a measure of the Information Leakage that the soft concept provided. **This is only valid because of the way we constructed our tree**, by first using the hard concepts to fit the global tree and then specialising the leaf nodes with soft concept splits.

### A.3 LEAKAGE INSPECTION: A NAIVE SOLUTION

A naive attempt to inspect any Information Leakage that the Soft Model exploited compared to a Hard CBM is to first train the same concept predictor for both models, and then use a separate Decision Tree classifier for each CBM as label predictor. In the first case, the inputs of the Tree are binary, ground truth concepts, whereas in the second case the inputs are the predicted concept probabilities of the concept encoder. To answer the question, we then need to compare two Trees. For reference, we visualise the Decision Trees trained for the Morpho-MNIST example Castro et al. (2018) with a constraint on the minimum samples per lead of $msl = 150$. The trees are shown side-by-side in Fig. 6.

We immediately observe that the task is not trivial because the structure of the two trees can be very different. Even if the root node uses the same concept in both trees to perform the primary split (e.g. "length xlarge"), the different threshold value used in each case may cause the resulting subsets to be completely different. Thus, the two trees may be incomparable after the first split, since the respective children nodes may use a completely different structure to distinguish their samples.

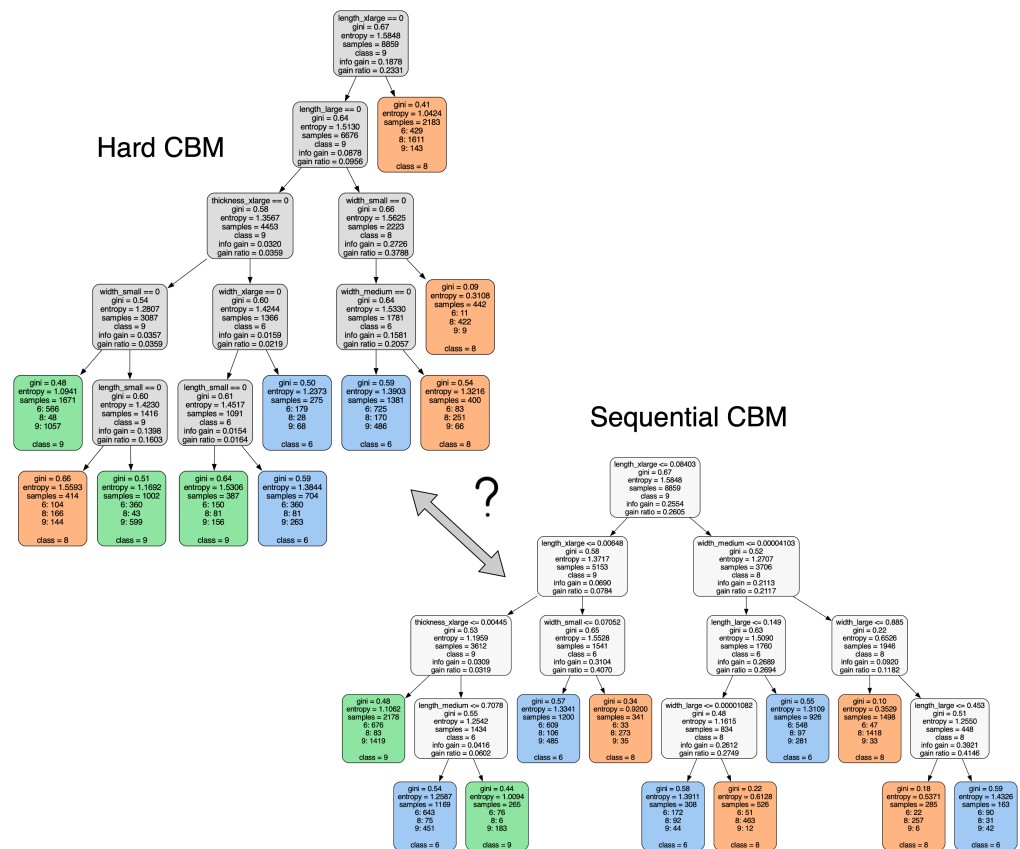

Figure 6: Leakage Inspection: Naive Comparison of a Hard and a Soft Sequential CBM.

## A.4 COMPLEXITY ANALYSIS FOR THE LABEL PREDICTOR OF MCBM-SEQ

An advantage of our approach is the limited computational overhead compared to training a standard independent CBM, because the concept encoder is trained only once. The individual Sequential CBMs all share the same concept predictor. Thus, the only overhead is that of training a sub-tree for each leaf of the global tree. In practice, this added computational cost is minimal because each tree only has access to a small subset $n_i$ of the total samples $n$, where $\sum_i^d n_i = n$ and $d$ is the number of leaf nodes in the global tree.

More specifically, the time complexity of a Decision Tree is $O(mn\,log_2 n)$ according to Sani et al. (2018), where $n$ is the total number of samples and $m$ is the number of attributes. This holds because the computational cost of performing a split based on one attribute follows the recursion formula of the divide and conquer algorithm $O(n\,log_2 n)$, and this process is repeated for all $m$ attributes to find the best split. Thus, the total cost $J$ of training the label predictor in the Leakage Inspection algorithm can be written as the cost of training the global tree and all its sub-trees:

$$J = J_{global} + \sum_i^d J_i \longrightarrow O(mn\,log_2 n) \tag{11}$$

The complexity is thus equivalent to that of a purely soft Sequential CBM that uses a single decision tree as label predictor, since the $msl$ constraint of the global and sub-trees is the same.

## A.5  MCBM-JOINT

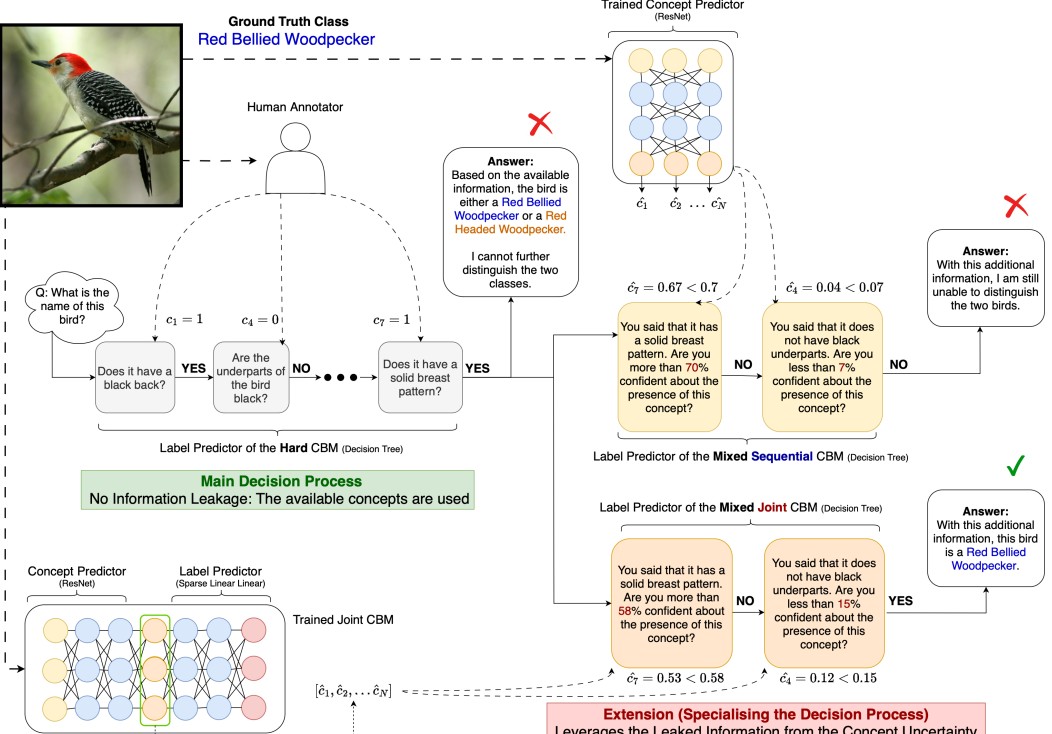

Figure 7: The Decision Making Process of The Mixed Joint CBM Algorithm (MCBM-Joint) when classifying an image with annotated concepts. The concept probabilities of the Joint CBM can be used to specialise the decision-making process of the label predictor when a) the available annotated concepts are not sufficient, and b) the concept probabilities of the Sequential CBM are also not sufficient (The probability numbers shown are constructed only for visualisation purposes).

The **Mixed Joint CBM algorithm (MCBM-Joint)** is a direct extension to MCBM-Seq and the underlying logic is presented in Fig. 7. For the given example with annotated concepts, both the Decision Tree of the Hard CBM and its path extension of MCBM-Seq cannot identify the class of the bird. If we also have in our disposal a trained Joint CBM, we could repeat the algorithm and find a new path extension using the concept probabilities from the concept predictor of the Joint CBM. This time, the updated decision path is able to make the correct distinction.

### A.6  DATASETS: ADDITIONAL DETAILS

**CUB**: The dataset is concept-complete, as all classes can be distinguished based on the ground truth annotated concepts at training time. To simulate the scenario of missing concept information, we select 45 out of the 112 concepts by picking all concepts from the following concept groups: "has-bill-shape", "has-wing-color", "has-upperparts-color", "has-underparts-color", "has-breast-pattern", "has-back-color", "has-upper-tail-color", "has-breast-color".

**MIMIC-IV**: From the total number of ICU admissions, we extract those patients with only one reported ICU Admission and whose age is between 18 and 90 years old. Moreover, for our mortality prediction task, we only select patients that were in the ICU for 48 hours or less. We use the following measurements as features for the length of their stay: creatinine, urine, norepinephrine, epinephrine, dobutamine, dopamine, mean blood pressure, P/F ratio, num. plalelets. Using these features, we calculate the SOFA scores Lambden et al. (2019), which provide an assessment for the condition of the following systems in the human body: respiratory, coagulation, liver (hepatic), cardiovascular, neurological (Glasgow coma scale) and renal. To construct concepts, we annotate

---

**Algorithm 2** Mixed Joint CBM Training (MCBM-Joint)

---

**Input:** $N = \{1, ..., n\}$ samples; $K = \{1, ..., k\}$ concepts; $R = \{1, ..., r\}$ classes;
   A dataset $D_{train} = \{X, C, Y\}$, where $X \subset \mathbb{R}^d$, $C \subset \{0, 1\}^k$, $Y \subset \{0, 1\}^r$;
   A set of $N_{test}$ samples with $X_{test} \subset \mathbb{R}^d$; Minimum Samples per leaf ($msl$).

**Output:** A hard tree: $T_{hard}$; a set of soft trees using concept probabilities from a
   Sequential CBM: $T_{seq} = \{T_1, ..., T_M\}$; a set of soft trees using concept
   probabilities from a Joint CBM: $T_{joint} = \{T_1, ..., T_M\}$; test predictions $\hat{Y}$.

1: **procedure** TRAINING($D_{train}, msl$)
2:     Train the concept predictor $f : X \to \hat{C}$, where $\hat{C} \subset [0, 1]^k$ (Eq. 1);
3:     Fine-Tune the predictor $f_{joint}$ using the Joint-Training objective of Eq .3;
4:     ...
5:     **return** $T_{hard}, T_{seq}, T_{joint}$
6: **end procedure**
7:
8: **procedure** TRAINING SUB-TREE($D_{train}, f, T_{hard}$)
9:     (Same as in Algorithm 1);
10:     **return** $T_{soft}$
11: **end procedure**
12:
13: **procedure** EVALUATION($X_{test}, f, T_{hard}, T_{joint}$)
14:     (Same as in Algorithm 1);
15:     **return** $\hat{Y}$
16: **end procedure**

---

SOFA scores 0-1 as concept 0 (normal), 2-3 as concept 1 (moderate), and 4 as concept 2 (severe). We thus have 18 concepts, 3 per system.

## A.7 HYPER-PARAMETER SETTING

We provide the hyper-parameters used for training the independent Concept and Label Predictors of MCBM-Seq. In the case of MCBM-Joint, an additional fine-tuning step of the concept predictor is performed using the Joint objective of Eq. 3. We test either the CHAID (Kass, 1980) or the CART (Breiman et al., 1984) algorithm to train the Decision Trees.

Table 4: Hyperparameter setting of MCBM-Seq, MCBM-Joint used by Morpho-MNIST, CUB-200 and MIMIC-IV

| Setting | Morpho-MNIST | CUB-200 | MIMIC-IV |
|---|---|---|---|
| Concept Predictor | LeNet | ResNet-18 | MLP: [128,64,2] |
| Learning rate ($x \to c$) | 0.001 | 0.001 | 0.01 |
| Optimiser | Adam | Adam | Adam |
| Weight-decay | 0.00001 | 0.00001 | 0 |
| Epochs | 200 | 300 | 50 |
| Decision Tree Algorithm used | CHAID | CART | CHAID |
| Minimum samples per leaf ($msl$) | 1 | 150 | 30 |
| Probability Calibration | Temperature | Platt | Temperature |
| Joint-training epochs | 50 | 100 | 50 |

## A.8 CONTROLLING THE GROUP SIZE

### A.8.1 MORPHO-MNIST

We test the size of the resulting trees compared to the performance accuracy for the Morpho-MNIST dataset. We observe that the **msl** constraint controls the Performance/Interpretability trade-off. As expected, both the number of Tree nodes and the Task Accuracy increase as the $msl$ constraint is reduced. However, an $msl$ close to 1 may not improve performance or even result in over-fitting. The advantage of the **msl** hyper-parameter is that it is human intuitive, as it directly controls the minimum size of our desired groups to perform explanations.

| **msl = 150** | **MCBM-Seq** | **MCBM-Joint** $\lambda_C = 100$ |
|---|---|---|
| Task Accuracy | 0.496 | 0.560 |
| Concept Accuracy | 0.894 | 0.905 |
| Num. Tree Nodes | 147 | 155 |

| **msl = 20** | **MCBM-Seq** | **MCBM-Joint** $\lambda_C = 100$ |
|---|---|---|
| Task Accuracy | 0.551 | 0.595 |
| Concept Accuracy | 0.894 | 0.905 |
| Num. Tree Nodes | 795 | 756 |

| **msl = 5** | **MCBM-Seq** | **MCBM-Joint** $\lambda_C = 100$ |
|---|---|---|
| Task Accuracy | 0.552 | 0.708 |
| Concept Accuracy | 0.894 | 0.905 |
| Num. Tree Nodes | 2455 | 2406 |

Table 5: Performance of MCBM-Seq, MCBM-Joint on Morpho-MNIST, with decreasing $msl$.

### A.8.2 MIMIC-IV

In this section we demonstrate that the $msl$ constraint does not only affect the Task Accuracy and the total size of the MCBM-Seq or MCBM-Joint tree, but also the meaning of our explanations in terms of the Information Leakage inspected by either method.

In the following figures, the MIMIC-IV trees are shown after the application of the MCBM-Seq algorithm for low, medium and high $msl$ values. The nodes annotated dark grey show the splits of the global tree, and the nodes with light grey show those of a leaky extension, if found. When $msl = 150$, we observe that the tree is too restricted for this small-sized dataset. Yet, the MCBM-Seq is able to find a leaky split under this constraint for patients whose renal system is in moderate condition (SOFA score 2-3). If the concept predictor is more than 80% confident that the condition of a given patient is indeed moderate, the label predictor is more confident that this patient may live. When the constraint is slightly relaxed at $msl = 70$, we observe that the corresponding tree can split the same group of patients with a second hard concept (condition of the cardiovascular system) and does not need to use Information Leakage. Thus, this constraint fits better for this dataset. For medium-sized groups, using an $msl = 50$, we observe that the predictor can fit the data even better, without the presence of leakage. However, when $msl = 30$, the two discovered leaky rules are very non-intuitive: When the concept predictor is more than 90% confident that the renal system is in severe condition, it is more likely that a patient may live. When a rule is very non-intuitive, it indicates a high noise in concept annotations. On the contrary, the leaky rule described in the first tree aligns with human intuition. This case-study shows that tuning the $msl$ constraint is critical for the performance and interpretability of our methods. Yet, this is the only single hyper-parameter that our label-predictor has.

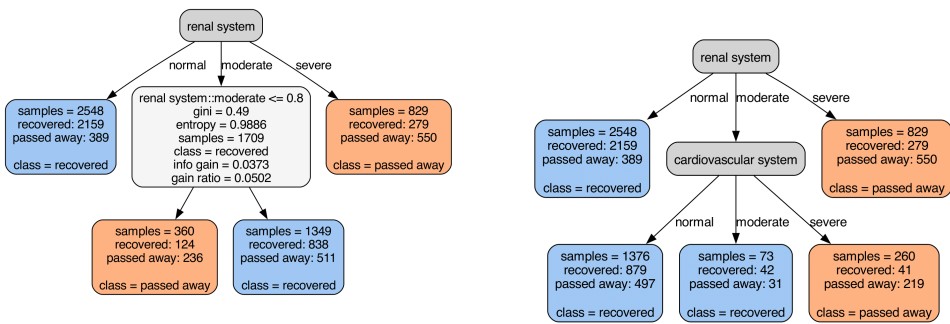

(a) MCBM-Seq on MIMIC-IV: $msl = 150$.  (b) MCBM-Seq on MIMIC-IV: $msl = 70$.

Figure 8: MCBM-Seq on MIMIC-IV with groups of large size.

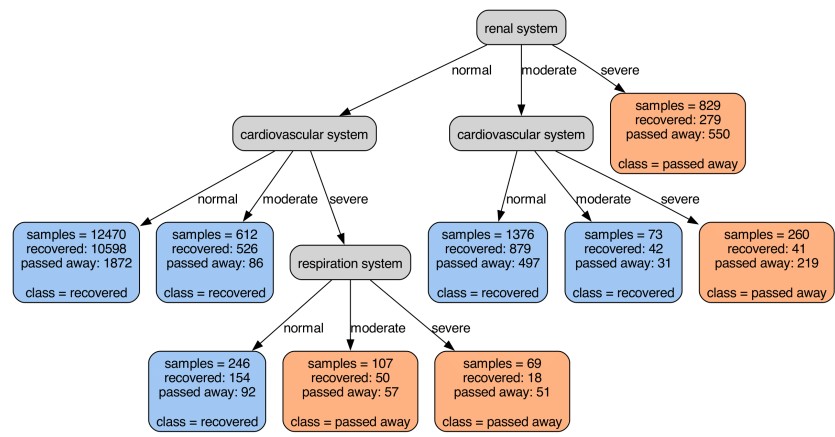

Figure 9: MCBM-Seq on MIMIC-IV with groups of medium size: $msl = 50$.

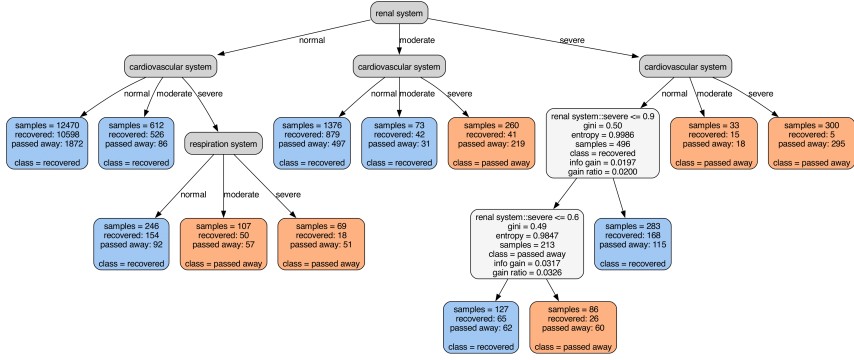

Figure 10: MCBM-Seq on MIMIC-IV with groups of small size: $msl = 30$.

## A.9  MCBM-SEQ ON MORPHO-MNIST: FULL TREES

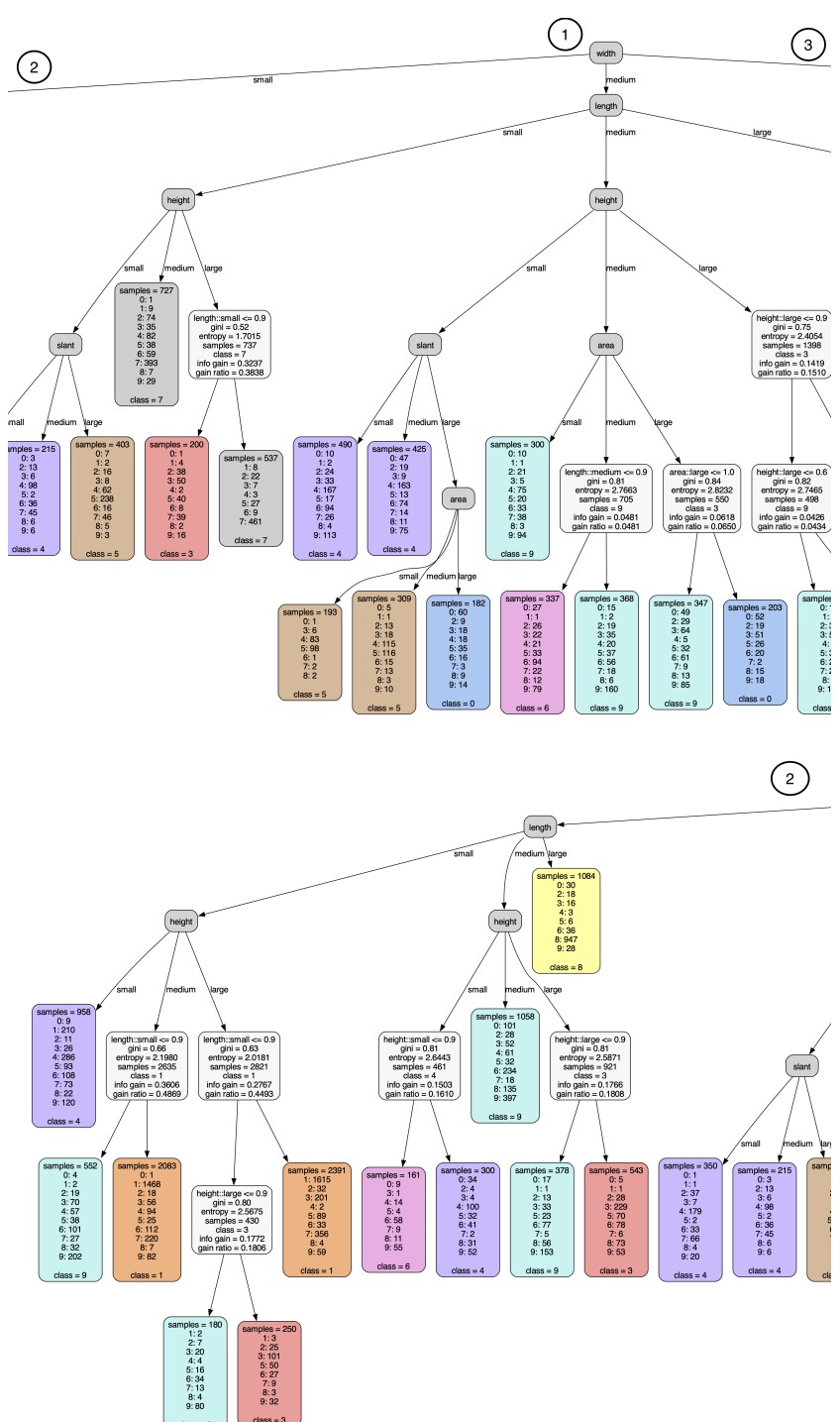

Figure 11: **PART 1**: MCBM-Seq - Full Merged Tree on Morpho-MNIST with $msl = 150$, split in four images. The dark grey nodes show the decision rules of the global tree, and do not introduce leakage. The white grey nodes show the leaky splits of the corresponding sub-tree, if found. Indicators in the top of the pictures indicate where the following or previous picture starts. Indicator 1 shows the root of the tree.

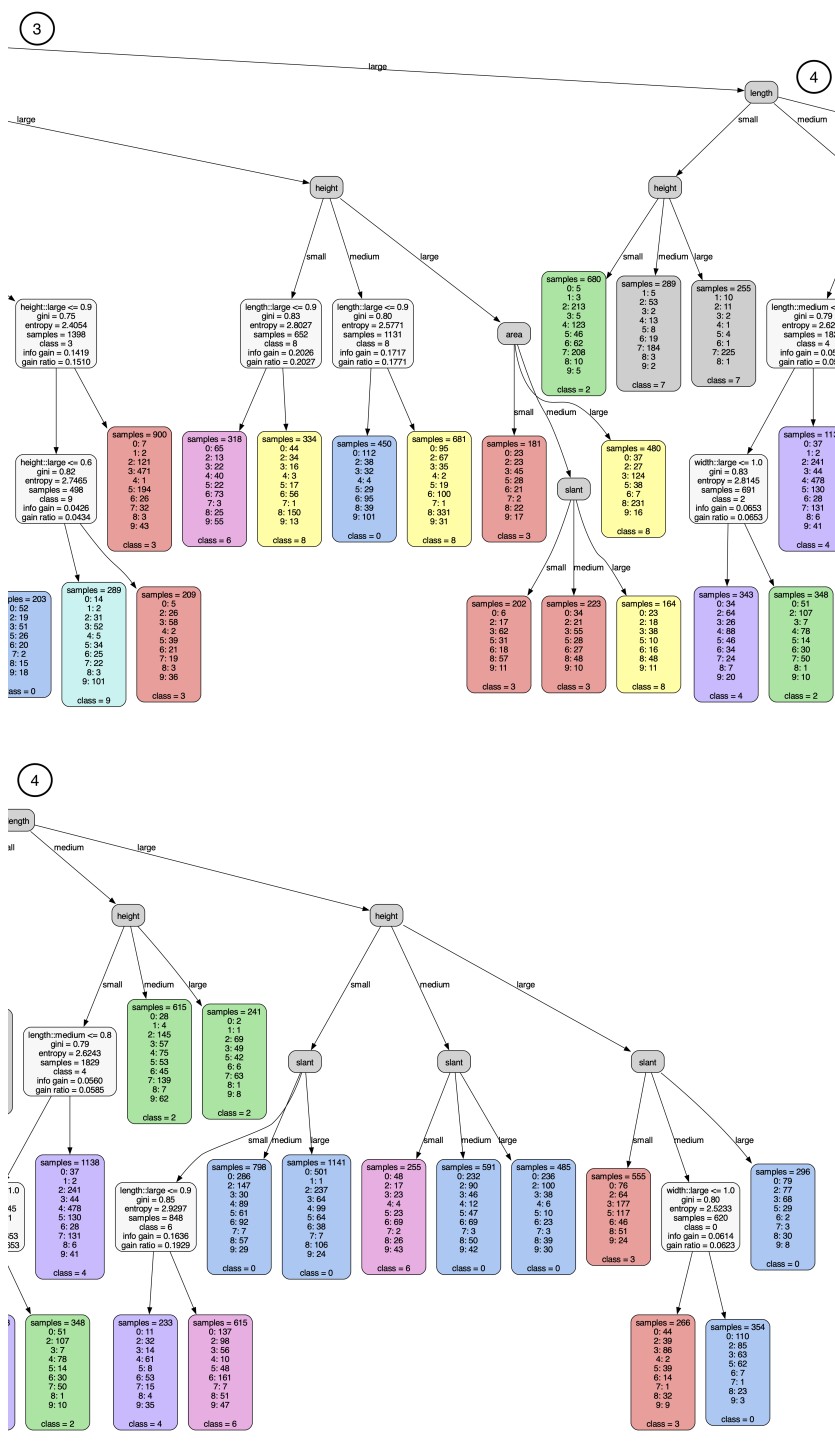

Figure 12: **PART 2**: MCBM-Seq - Full Merged Tree on Morpho-MNIST with $msl = 150$, split in four images. The dark grey nodes show the decision rules of the global tree, and do not introduce leakage. The white grey nodes show the leaky splits of the corresponding sub-tree, if found. Indicators in the top of the pictures indicate where the following or previous picture starts. Indicator 1 shows the root of the tree.

### A.9.1 DECISION PATHS

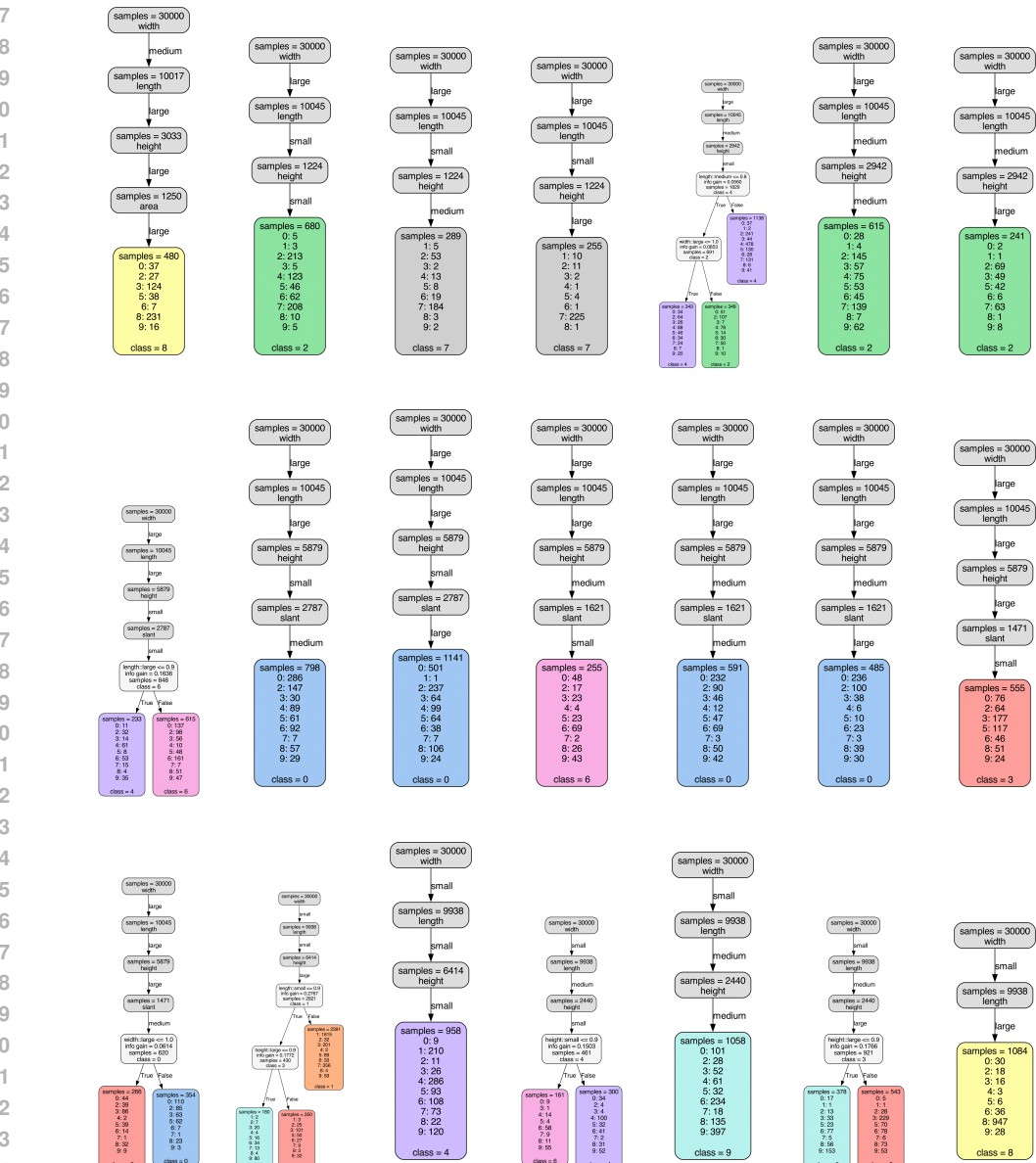

Figure 13: **PART 1**: MCBM-Seq - Full Merged Tree on Morpho-MNIST with $msl = 150$: All decision paths of the merged tree. If a sub-tree is found for a decision path, the complete sub-tree is shown.

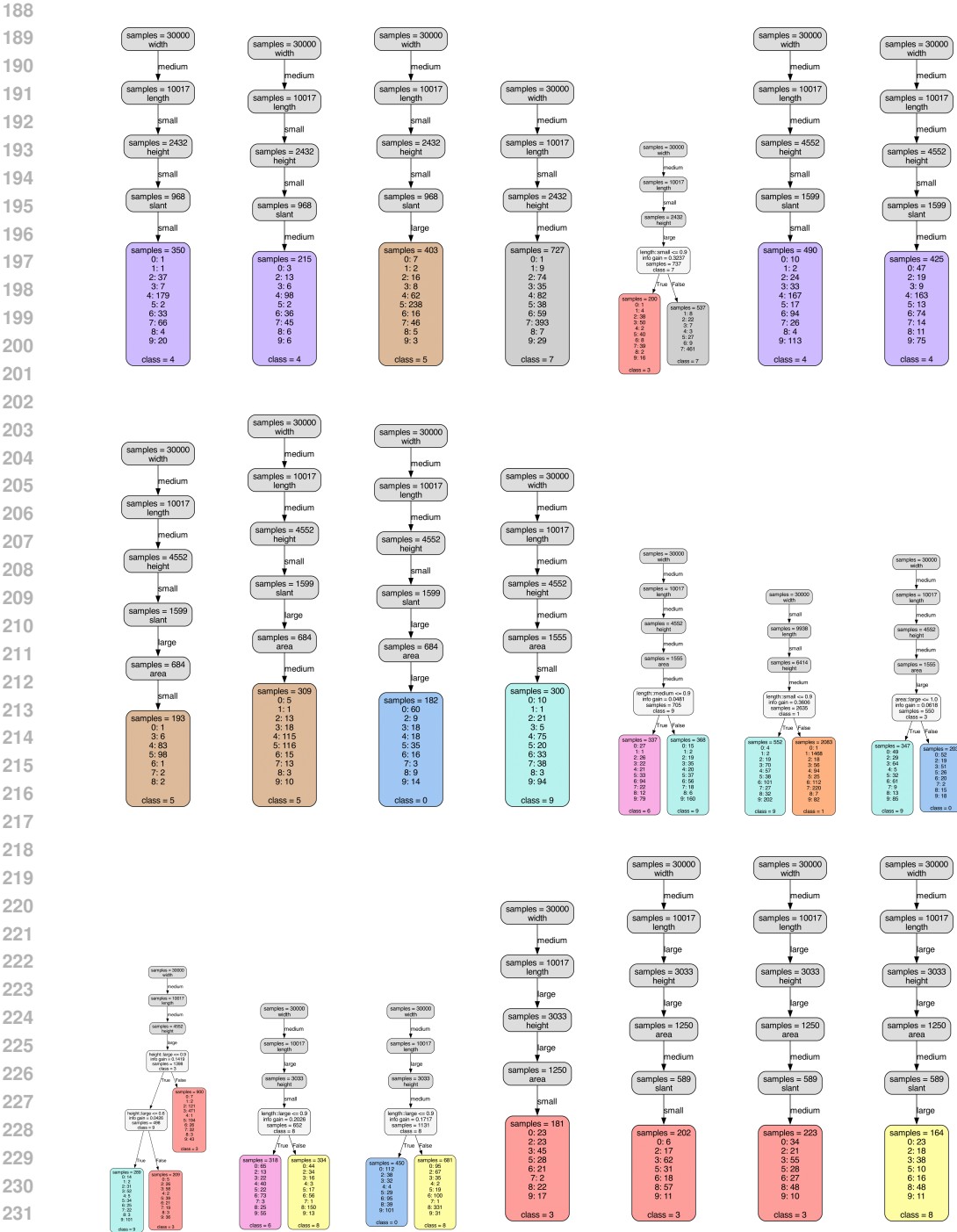

Figure 14: **PART 2**: MCBM-Seq - Full Merged Tree on Morpho-MNIST with $msl = 150$: All decision paths of the merged tree. If a sub-tree is found for a decision path, the complete sub-tree is shown.

## A.10   CUB

We give the full decision path of the case study described in section 5.3 in Fig. 15 below. As we described, the label predictor observes that the concept predictor is **less than** 70% **confident** that the Red Bellied Woodpeckers have a solid breast pattern, while it is more confident for the vast majority of Red Headed Woodpeckers that they possess this attribute. While it is not the main focus of this work, we provide a visually plausible intuition for this result in Fig. 16, by examining many birds of the two classes. Our intuition is that the concept predictor generally assigns higher probabilities to Red Headed Woodpeckers because their breast colour is completely white and thus their pattern is solid, while that of Red Bellied Woodpeckers often shows orange dots. This is an example which shows that calibrated input probabilities can often be human-intuitive.

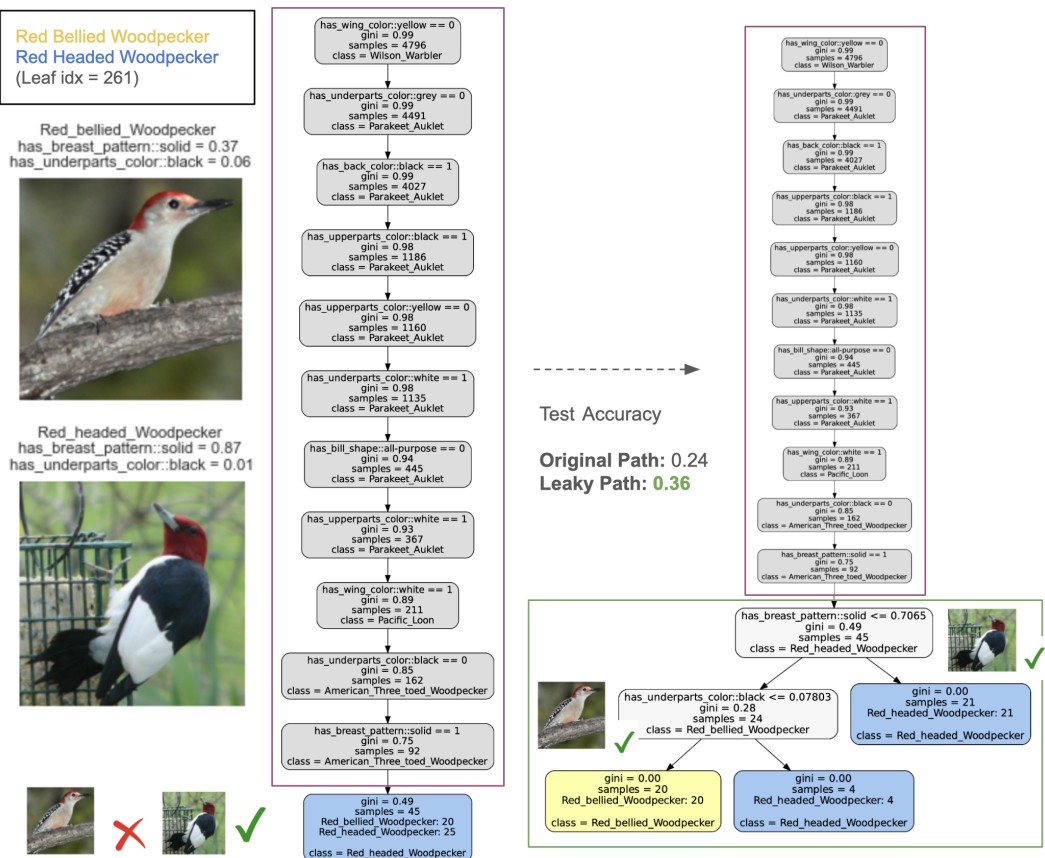

Figure 15: A case study in the CUB Wah et al. (2011) dataset. The Figure shows the corresponding decision path on the global tree (left) and how this is extended using Information Leakage (right) by the MCBM-Seq method to improve performance.

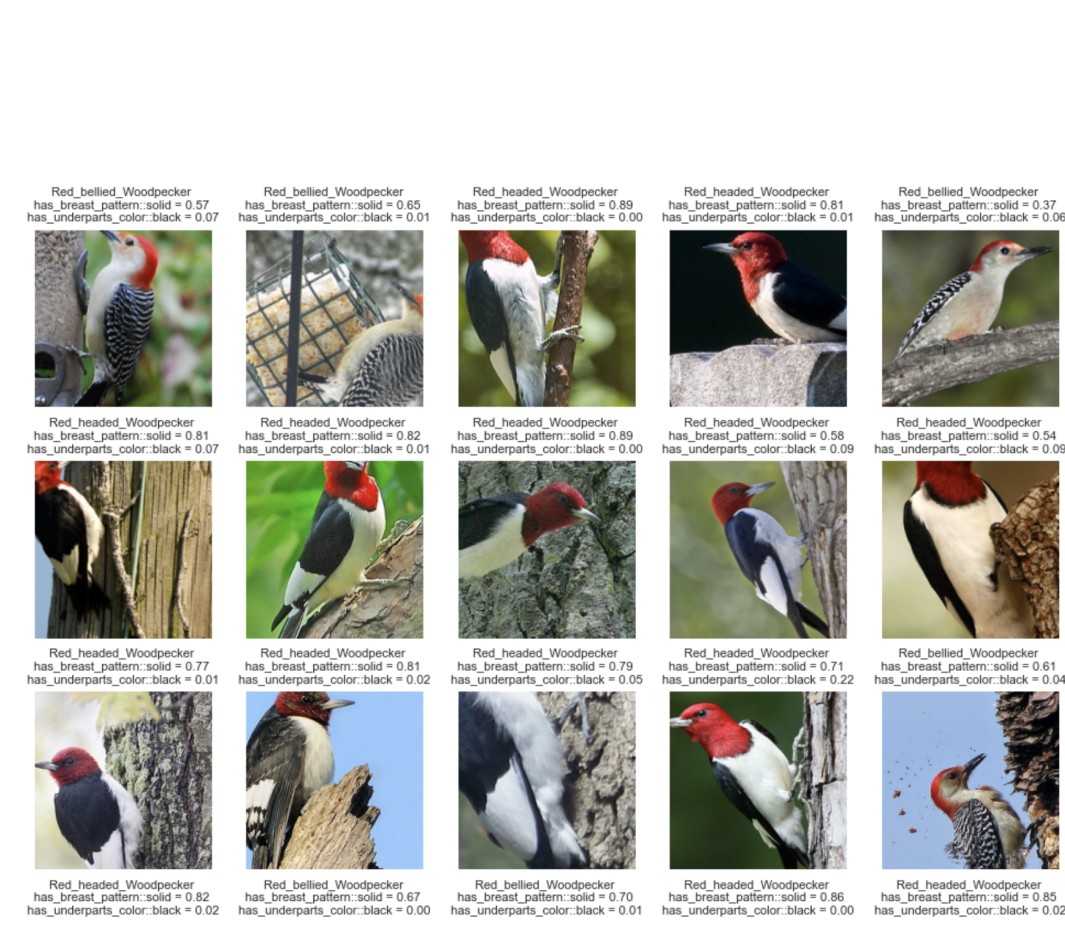

Figure 16: CUB: More birds of the classes "Red bellied Woodpecker" and "Red headed Woodpecker". The confidence of the concept predictor is shown per bird for the concepts: "has-breast-pattern-solid", and "has-underparts-colour-black".

### A.10.1 CUB:MORE DECISION PATHS

Figure 17: Indicative Decision Paths of the merged MCBM-Seq tree on CUB. For the selected paths, a leaky sub-tree is found. Each new decision path distinguishes a pair of bird classes that were previously indistinguihable.

