# OpenReview forum: "Controlling Information Leakage in Concept Bottleneck Models with Trees"
_ICLR.cc/2025/Conference — ICLR 2025 Conference Withdrawn Submission_

### Official Review · Reviewer_VU8H · 2024-10-20

**Soundness:** 3
**Presentation:** 2
**Contribution:** 2
**Rating:** 3
**Confidence:** 4

**Summary:**

The authors study a particular form of information leakage in decision-tree CBMs. They introduce a metric and method to quantify this information leakage by comparing the decision-tree paths of hard CBMs with their soft counterparts. Their method induces little to no drops in task / concept accuracy.

**Strengths:**

- the authors study the interesting and important problem of how to build trustworthy CBMs
- the authors perform experiments on diverse datasets

**Weaknesses:**

- the main issue seems to be a lack of rationale for definition 3.1, which forms the basis for the paper. The authors quantify *information leakage* as the amount of “unintended information” that is used to predict the label with soft concepts that is not present in hard representation. Why captures whether this information is “unintended”. It seems to me that the author’s information measure needlessly penalizes soft concepts that provide extra, intended information. For example, in Morpho-MNIST, continuous features  (i.e. thickness, area, length, width, height, and slant of digits) are binned and then used for prediction. Soft concepts may improve the model by removing the binning, making these features more reliable.
- It is also unclear how quantifying this information leakage is useful. It would be nice to see whether this information could be used, e.g. to improve concepts for a downstream task or in a human user study where concepts are shown to be more understandable. This is especially important seeing as the introduced method seems to slightly decrease some desirable metrics (Table 1)

**Questions:**

- why do the authors focus on decision trees (e.g. as opposed to a linear model CBM)? The paper only compares against Entropy Net & a black-box baseline
- minutia
    - line 141 “linear layer decision trees” - do the authors mean “linear layer or decision trees”
    - line 144 “networks f and g” - is g a network? line 141 would suggest it is not
    - why is this a reasonable definition?
    - line 189 “as the trees be incomparable”

---

> ### Author Response · Authors · 2024-11-19
> **Author Response**
>
> We sincerely thank the reviewer for reading our work and providing comments. We will plan to respond to the highlighted weaknesses and questions in a sequential manner.
>
> **Weakness 1**: *"the main issue seems to be a lack of rationale for definition 3.1, which forms the basis for the paper. ... more reliable."*
>
> **Response**: First, we would like to clarify that our method is designed especially for categorical concepts, since the vast majority of real-world concept-annotated datasets have categorical and not continuous concepts, including CUB and MIMIC which are explored in this paper. Morpho-MNIST is an exception and has continuous concepts, which we could have indeed leveraged differently. However, in our case this particular dataset is used as a toy example to demonstrate how are method works. Thus, we perform the binning operation to create the “small”, “medium” and “large” categorical concepts.
>
> We would also like to clarify again the definition of a “hard” and a “soft” categorical concept, following the existing formulation of [1]. A hard categorical concept is a boolean concept, where “1” indicates its presence and “0” indicates its absence. A soft categorical concept refers to a concept probability, indicating the confidence of the concept predictor about its presence. Thus, a “soft” concept is different from a continuous concept.
>
> [1] Havasi, M., Parbhoo, S., & Doshi-Velez, F. (2022). Addressing leakage in concept bottleneck models. Advances in Neural Information Processing Systems, 35, 23386-23397
>
> **Weakness 2**: *"It is also unclear how quantifying this information leakage is useful....metrics (Table 1)"*
>
> **Response**: We believe that quantifying leakage is a very important tool for analyzing the interpretability of Concept Bottleneck Models.
>
> To make this more clear, let us first adapt a practical example of leakage in CBMs described in [1]. Assume we have a CBM performing animal classification. When the concept predictor recognizes an image as a dog, it may predict a slightly higher likelihood to the concept ‘tail’ compared to an image of a cat. Let us assume for example that the concept predictor gives a likelihood around 0.8 to the majority of dogs for the concept ‘tail’, and around 0.6 for the majority of cats. The ground truth concept ‘tail’ for both classes is 1 (meaning that both animals have a tail). Even though cats and dogs are indistinguishable based on this ground truth concept, the label predictor may still compare the two soft concept probabilities (0.6 and 0.8) and make an accurate classification. However, is this concept-based explanation useful? The label predictor relies on a difference in likelihoods that may (or may not) be interpretable to a human. For example, it might be the case that in most images of cats their tail is not clearly shown due to the orientation of the cat, whereas in most images of dogs the tail is clearly visible. This phenomenon is called “leakage” because unintended information from the input is captured in the concept likelihood (the orientation of cats and dogs). Thus, the concept predictor assigns a higher likelihood to dogs having this concept. However, this may be just an assumption. Regardless of whether this difference is intuitive or not, we believe **it is crucial to first find a systematic way to isolate instances where the label predictor takes advantage of such differences**, in order for us to then proceed to further analysis. In our paper, a similar case study is performed for the Woodpecker example of Figure 2.
>
> **Thus, our work first provides a systematic way to identify those instances affected by leakage using our tree structure**. Specifically, we first train a tree using only the ground truth concepts, which we name the “global tree” (step 1 in Figure 1). Each leaf node in the global tree corresponds to a subset of examples that are indistinguishable based on their ground truth concepts. For example, if we observe the leftmost decision path in the toy MNIST example of Figure 3, there is a leaf node corresponding to group of 926 digits (278 “6”s, 33 “8”s and 615 “9”s), and all these digits have small length and small thickness. We see that this group was not extended with leakage after performing our algorithm, i.e. a sub-tree was not found (step 2 in Figure 1). This means that the label predictor could not take advantage of any differences in the likelihood of concepts “length:small” and “thickness:small” to further split this group. On the other hand, the third decision path from the left, which corresponds to the group of digits having small length and large thickness, was extended with two decision rules corresponding to differences in concept likelihoods (leakage): “length:small < 0.8” and “thickness:large < 0.8”.
>
> *(The response continues in the next comment)*

---

> ### Author Response · Authors · 2024-11-19
> **Author Response (Continue)**
>
> *(continuing the previous response...)*
>
> Identifying such instances can then assist our decision-making when deriving concept-based explanations per decision path. This corresponds to the analysis of section 5.3. For better clarity, we update this section by defining two scenarios:
>
> * *If a group (leaf node) **cannot** be further split using leakage, i.e. a sub-tree is not found*: This shows that the particular group is not affected by information leakage, which is desirable. In addition, if the task accuracy for this particular group is high, we may consider this an ideal classification example, because the available concept annotations are sufficient for an accurate distinction. The fact that we can identify and isolate such groups is highlighted as a key advantage of our work. **Unlike a purely soft CBM, leakage will not impact these groups, and thus the concept explanations for those groups are both leakage-free and accurate**. If, on the other hand, the task accuracy is not sufficient, we may flag this group to an expert to either annotate additional concepts or perform an intervention (future work).
>
>
> * *If a group (leaf node) **can** be further split using leakage, i.e. a sub-tree is found*: Then this group can be flagged for additional analysis. We describe a detailed case study for the Woodpecker example in page 10, paragraph: “Our tree-structure allows for meaningful group-specific explanations”. We urge the user to investigate if this difference in likelihoods (leakage) might be intuitive or not, similar to the cat-dog example we described in the beginning. Then the user has the following options: a) rely solely on the decision process of the global tree to derive a perfectly understandable, leak-free explanation, b) extend the decision process with the sub-tree of MCBM-Seq if this likelihood difference seems intuitive, c) use MCBM-Joint’s less intuitive probabilities for maximum accuracy, d) flag this group to an expert to either annotate additional concepts or perform an intervention (future work).
>
> In conclusion, identifying leakage is useful because it provides these analysis tools to a decision maker when deriving explanations, which are not available to a standard CBM. We currently develop future work providing interventions and concept discovery strategies specifically when leakage is observed. However, future leakage mitigation strategies first require a method that controls and inspects leakage for specific sub-groups, thus we believe this work may be used as a very useful analysis tool while not sacrificing the task performance of a standard CBM (the task performance is comparable).
>
> **Question**: *Why do the authors focus on decision trees (e.g. as opposed to a linear model CBM)? The paper only compares against Entropy Net & a black-box baseline*.
>
> **Response**: We focus on decision trees because they offer us the advantage of inspecting leakage, which is not provided by a linear model or a neural network. More specifically, they allow us to inspect leakage defined in Eq. (4) using our formulation of Eq. (5) and Appendix A.2. Quantifying Leakage is not straightforward using a linear model or the Entropy-Net, i.e. it is not evident how the mutual information $I(y; \hat{c}|c)$ in Eq. (4) could be approximated using such models in a similarly efficient way. Moreover, trees allow us to inspect leakage for specific groups and derive group-specific explanations in the form of decision paths by controlling the “minimum samples per leaf (msl)” constraint, as highlighted in section 5.3. In contrast, linear models as well as the Entropy-Net model only allow us to form either instance-specific or class-specific explanations. In terms of task and explanation accuracy, we compare our method directly with the Entropy-Net model because it is currently the state-of-the art method for concept-based explanations, outperforming simple linear models [2].
>
> [2] Barbiero, P., Ciravegna, G., Giannini, F., Lió, P., Gori, M., & Melacci, S. (2022). Entropy-Based Logic Explanations of Neural Networks. Proceedings of the AAAI Conference on Artificial Intelligence, 36(6), 6046-6054.

---

> > ### Comment · Reviewer_VU8H · 2024-11-25
> >
> > I thank the authors' for their response (and encourage them to incorporate these notes in the manuscript), but maintain my rating.

---

### Official Review · Reviewer_MJtJ · 2024-10-30

**Soundness:** 3
**Presentation:** 3
**Contribution:** 3
**Rating:** 5
**Confidence:** 3

**Summary:**

Concept bottleneck models (CBMs) is a valuable technique for enhancing the interpretability and explainability of deep learning models; however, they have recently been shown to suffer from information leakage issues. This work proposes a new decision tree-based method to address the leakage issue. Unlike the original CBM which used a small network (e.g. a linear model) to predict the label Y from the (soft or hard) concept representation C, this work uses a decision tree as the predictor. The authors smartly show that by comparing the cases when  soft concepts and hard concepts are used to construct the tree, it is possible to inspect and control information leakage corresponding to specific configurations of concepts. The method is evaluated on several standard datasets.

**Strengths:**

- The paper introduces a new decision tree-based method for quantifying and controlling information leakage in CBMs, which to the best of my knowledge is very novel. I personally think that the use of tree-based model in CBMs itself may already be a good contribution before its applications in addressing information leakage.
- The writing is mostly clear and easy to follow. I especially appreciate Figure 1 which helps readers to intuitive understand the main idea and pipeline of the proposed methodology;
- To the best of my knowledge, this is the first work to formulate information leakage issues in CBMs. For this purpose, the authors propose a general, information-theoretic metric (Definition 3.1). The applicability of this metric is well beyond the specific method considered in this paper (tree-based CBMs);
- The work also offers a new method for inspecting how information is leaked and used in a more fine-grained and interpretable manner. By inspecting each path in the decision tree, practitioner can understand how hard concepts are insufficient for accurate predictions and how additional information could enhance these predictions (thereby leads to information leakage). This interpretability, facilitated by the use of decision tree, is a notable innovation over existing methods;
- Reproducibility: the authors have provided certain implementation details as well as offering anonymous code repo.

**Weaknesses:**

- (Major) There might be, in my opinion, a potential discrepancy between the information leakage metric defined (Definition 3.1) and the actual tree-based implementation for computing this information leakage (see the “questions” section below);
- (Major) Related to the above point, it seems that the information leakage problem addressed in this paper differs slightly from that studied in existing works. Specifically, this work appears to focus on computing information leakage corresponding to specific configurations of hard concepts $C$ i.e. $I(\hat{C}; Y|C=const)$ (where $C$ equals to some constant) rather than calculating $I(\hat{C}; Y|C)$ (where both $\hat{C}$ and $C$ are random variables);
- (Major) While the insights and ideas presented are highly novel, the proposed three-phase method also seems more complex than existing approaches aiming to address information leakage [1, 2, 3];
- (Minor) The work has not been compared to state-of-the-art methods for addressing information leakage, such as CEM [1], PCBM [2] and [3];

Further comments: it seems more natural to rename the method to "Tree-based CBM". This is just a recommendation for consideration.

*References*

[1] Concept Embedding Models. NeurIPS 2022

[2] Post-hoc Concept Bottleneck Models. NeurIPS 2022

[3] Addressing leakage in concept bottleneck models. NeurIPS 2022




*Disclaimer: the reviewer is not the author of any of these papers.

**Questions:**

- The information gain you computed in eq.(5) seems to condition on a particular configuration of the hard concepts (e.g. ck = [0, 1, 0]) corresponding to the leaf node. Is this correct?
- How is the calibration process described in lines 211-241 actually performed? It may not be immediately clear to those who are unfamiliar with the specific calibration technique mentioned. The author could consider to include a short description of these processes in future refinement.
- The work has not been compared to other methods for addressing information leakage in CBMs e.g. [1, 2, 3]. Could the authors provide a justification for this omission?

---

> ### Author Response · Authors · 2024-11-19
> **Author Response**
>
> We sincerely thank the reviewer for reading our work and providing comments. We will plan to respond to the highlighted weaknesses and questions in a sequential manner.
>
> **Weaknesses 1-2 and Question 1**: *"(Major) There might be, in my opinion, a potential discrepancy between the information leakage metric defined (Definition 3.1) ... (Major) Related to the above point, it seems that the information leakage problem addressed in this paper differs slightly from that studied in existing works..."*
>
> **Response**: Thank you for this great question. You are correct. The definition 3.1 we provide is general and indeed refers to $\hat{C}$ and $C$ being random variables. Indeed, we calculate the metric **per leaf node** and **per decision split** by conditioning the random values $\hat{C}$ and $C$ appropriately. Refer to Appendix A.2, page 14 for the full description. We indeed denote $I_{Leakage}$ as $I_{Leakage}( \hat{c}_k )$ in Eq. 8, showing that this is the Information leakage induced by the specific split. We do not estimate the mutual information of definition 3.1 specifically, since we observed that it is more useful in practice to quantify leakage in specific groups rather than providing a global leakage estimate, as shown in the per-path analysis of section 5.3. We believe that conditioning on the random variables does not introduce a discrepancy, but rather makes this information metric more specific.
>
> **Weaknesses 3-4 and Question 3**: *"The work has not been compared to state-of-the-art methods for addressing information leakage, such as CEM [1], PCBM [2] and [3];"*
>
> **Response**: The three cited papers are included in our section 2 “Related Work”, with a short justification of why we believe they do not sufficiently address information leakage. Here, we elaborate for each one:
>
> * *“Concept Embedding Models. NeurIPS 2022”* [1]: This paper introduces the idea of a “concept embedding”, which was later adopted by more CBM papers such as PCBM [2]. In our work, we use scalar-valued concepts instead of concept embeddings, following the original CBM paper*. While concept embeddings achieve excellent task performance because they lead to more expressive concept representations, the authors do not comment on information leakage, i.e. they do not provide a justification about whether these concept embeddings also capture unintended (“leaked”) information from the inputs to improve the task performance. Instead, they propose the Concept Alignment Score (CAS) to measure how much learnt concept embeddings can be trusted as faithful representations of their ground truth concept labels. Their intuition is that clustering samples based on a faithful concept embedding would result in coherent clusters. While they show that CEMs achieve high CAS scores, we argue that their method may not be sufficiently interpretable because: i) They achieve CAS scores of around 80% in certain datasets, such as CUB and CelebA, which may imply the presence of leakage. ii) This approach does not indicate which subsets may suffer from the imperfect concept alignment, or how does this imperfection affect concept-based explanations. In contrast, our tree-based method allows for group-specific leakage examination in the form of decision paths, and gives the exact decision rules based on leakage. iii) The information captured in a high-dimensional concept embedding is unintuitive compared to a scalar-valued concept, which directly represents the probability (confidence) of the concept predictor.
>
> * *“Post-hoc Concept Bottleneck Models. NeurIPS 2022”* [2]. Similar to CEMs, this work uses concept embeddings but in the form of concept activation vectors (CAVs). Also, the authors do not address the issue of concept faithfulness or that of leakage, since they rely on multi-modal models to learn concepts that may be not annotated. While they effectively deal with the problem of missing concept annotations, their concept quality and faithfulness relies on the fidelity of their multimodal model.
>
> *(The response continues in the next comment)*

---

> ### Author Response · Authors · 2024-11-19
> **Author Response (Continue)**
>
> *(continuing the previous response...)*
>
> * *“Addressing leakage in concept bottleneck models. NeurIPS 2022”* [3]. This work is closer to our method, in the sense that a) it uses scalar-valued concepts and b) specifically addresses leakage. However, as we mention in our related work, “Havasi et al. (2022b) tackle missing information with a side channel and an auto-regressive concept predictor, but these approaches struggle with interpretability and disentanglement of residual information (Zabounidis et al., 2023)”. In more detail, all works that use a residual layer or side-channel (including this one) aim to let missing concept information pass directly from inputs to targets, letting the ground truth concept representations intact and not influenced by leakage. Yet, Zabounidis et al., 2023 highlight that the residual information is not guaranteed to capture this intended missing information, and the two representations may be entangled. We argue that the lack of transparency in the residual channel does not make these methods convincing enough for the specific problem.
>
> In conclusion, while we understand that the three-phase method may seem more complex, we argue that it is more effective and provides some novel advantages compared to existing methods, such as the ability to perform group-specific leakage examination in the form of decision paths and to identify the exact decision rules based on leakage. Thus, our work cannot be quantitatively compared with these previous works. Moreover, we argue that the three-phase method is not that complex in practice, because essentially it only involves the training of one global tree and individual sub-trees for the leaf nodes of this tree, along with an independently trained concept predictor like in all CBMs.

---

> ### Comment · Reviewer_MJtJ · 2024-11-25
>
> I admire the authors' effort to address my and other reviewer's concerns. All my questions have been answered, and some of the issues raised have been addressed. However, the work still lacks a direct empirical comparison to existing works for controlling information leakage (e.g. PCBM and CEM). At the same time, I still feel the three-stages unnecessarily heavy, especially that there is  currently no clear (empirical) evidence showing the advantages against existing methods. All these make me uncertain whether the work is borderline acceptable or not.
>
> I thereby maintain my current scores at this stage, and will determine my final score after discussing with other reviewers. My current score should be interpreted as a 5.5.
>
> Once again, thank your for your noticeable efforts in improving the work. Wish you the best of luck with the submission.

---

### Official Review · Reviewer_MYyQ · 2024-11-03

**Soundness:** 1
**Presentation:** 3
**Contribution:** 2
**Rating:** 5
**Confidence:** 4

**Summary:**

This work introduces Mixed Concept Bottleneck Models (MCBMs), a predictive model and an inspection tool that uses decision trees to analyze and control information leakage in traditional Concept Bottleneck Models (CBMs). By exploiting a hard independent CBM whose label predictor is a decision tree, this model constructs a CBM by expanding each of the original decision tree’s leaf nodes using new sub-trees that operate on soft concept representations for the concepts that are used to reach that node. This allows MCBMs to properly quantify leakage across each decision path and rule, producing group-based explanations. This paper evaluates MCBMs across three datasets and shows that they may lead to high-fidelity and interpretable explanations whilst performing similarly to equivalent hard, independently trained CBMs.

**Strengths:**

Thank you so much for submitting this work! I enjoyed reading this paper and learned a lot while reading it. Below are what I believe are this paper’s main strengths:

1. **[Originality] (Critical)** Introducing decision trees to capture, disambiguate, and control leakage in vanilla CBMs is, to the best of my knowledge, certainly novel. I like the general idea and believe others may also find it interesting and new.
2. **[Significance] (Major)** If shown to lead to actionable conclusions, I do believe that MCBM may have some impact as an analysis tool (more so than as a predictive model; see weaknesses below). Moreover, the simple yet useful formalization of information leakage (i.e., equation (4)), which MCBMs can very easily estimate, is a helpful step towards better understanding and studying information leakage. Therefore, this approach may be useful to others working in this space, particularly those interested in leakage. As such, I believe both of these contributions are potentially useful for the overall XAI community.
3. **[Quality] (Minor)** This work is well-placed within the concept-based XAI literature and does a good job of connecting MCBMs to existing work outside of this paper.
4. **[Clarity] (Major)** The paper is very well-written, easy to follow, and full of visual aids that truly help with its understanding.

**Weaknesses:**

In contrast, I believe the following are some of this work’s limitations:

1. **[Significance] (Critical)** I can see how the proposed approach may be useful for studying leakage in vanilla CBMs. However, I think a very strong case can be built against using MCBMs as a model for prediction in real-world cases, given their significant drops in performance compared to existing even simple baselines (e.g., joint CBMs). A case can be built to use a model that offers more interpretability (however one defines that) if the hit on performance is not very significant. In this case, however, the presented evidence suggests this hit can indeed be quite significant. Moreover, it is unclear how MCBMs compare to and how they could be used to analyze much more modern approaches (e.g., Post-hoc CBMs, CEMs, ProbCBMs, Energy-free CBMs,  etc.), all of which are much better than vanilla CBMs. As such, I have some doubts about the potential impact of this work without further evidence or contributions showing their use for modern baselines/pipelines/frameworks. See below for further questions on this particular topic, as this is my biggest concern/hesitation regarding this work.
2. **[Significance] (Critical)** Related to my concern above, although MCBM is claimed to be helpful not just as a model but also as an analysis tool, the current experiments fail to provide evidence that any conclusions extracted by analyzing MCBM’s outputs do indeed lead to actionable changes that improve the analyzed model. My current concern is that this work presents MCBM as both (1) a model and (2) an analysis tool, without providing sufficient evidence, in my opinion, that it leads to actionable significant improvements in either of those two directions.
3. **[Significance/Quality] (Major)** Concept interventions, a standard evaluation procedure for CBM-like models, are not evaluated anywhere in this work. As such, it is hard to fully understand the benefits of MCBMs over other existing baselines in this field.
4. **[Quality] (Major)** Against common good practices, no error bars are provided for any of the results. This makes it very difficult to judge for significance, and it is particularly important here as some of the gains are small enough that they could just be from noise.
5. **[Quality/Clarity] (Minor)** It is unclear how some of the key hyperparameters (e.g., $\texttt{msl}$) were selected for the different tasks. Given how sensitive MCBMs are shown to be to this hyperparameter (in the appendix), it is very important to verify that this hyperparameter was properly selected without accidental test-set leakage.

**Questions:**

**[Post rebuttal update: Changed my score to a *5: marginally below the acceptance threshold*]**

Currently, given some of my concerns with this work's framing and evaluation, and considering them w.r.t. the strengths I listed above, I am leaning towards rejecting this paper. However, I am absolutely happy to be convinced that some or all of my conclusions are wrong and to change my recommendation based on a discussion with the authors. For this, the following questions could help clarify/question some of my concerns:

1. **(Critical)** I understand that “trustworthy”/leakage-free explanations are always better. However, in the case where the concept set is incomplete (which is likely to be the case for any real-world dataset), what is the argument for using something like MCBM over any of the baselines that can achieve high accuracy in these setups (e.g., joint logit CBMs, Hybrid CBMs, Hybrid Post-hoc CBMs, CEMs, etc)? The performance difference between MCBM-Seq and joint CBMs, arguably a weak baseline for incompleteness compared to more recent approaches, seems to be large enough that one could construct a very convincing argument that any gains in reductions in concept leakage are not worth it in practice  (e.g.,  up to 25% absolute drop in task accuracy in CUB according to Table 2). This is my largest concern with this work. Am I misunderstanding something here? If not, what is the case for using something like an MCBM over any other existing approaches in practical scenarios where concepts are almost certainly bound to be incomplete? The reason why I am fixating on this is that there are several claims in the paper (e.g., “these tree-based approaches …achieve better accuracy on datasets with incomplete concept information” in Section 6) that do not appear to be backed by the same evidence presented in this paper. As such, assuming I correctly understand this work (happy to be convinced that I do not), I think these claims should be revised to represent better what the evidence shows.
2. **(Critical)** Related to the question above, if the interest is to use MCBM as a predictive model, do you have a sense of how MCBMs perform against any (not necessarily all) of the many high-performing modern baselines (CEMs/Post-hoc CBMs/ProbCBMs/Energy-based CBMs/etc)?
3. **(Critical)** If the argument to be built is that MCBMs are better for post-hoc analysis than prediction, then I would say the paper should focus more on the analysis part and show how MCBMs can be used to lead to actionable changes/edits/insights that improve the underlying model somehow. Section 5.3 shows some of this but falls short in that it does not convincingly show that some of the conclusions made from the MCBM’s analysis can lead to changes to the underlying model that indeed improve it under some intended metric. Therefore, do you have any empirical evidence of actionable changes derived from insights from MCBM that led to a model’s update improving its performance under a reasonable metric? Could these sorts of studies be extended to more modern architectures like those discussed above?
4. **(Critical)** Could you please provide error bars for the results in all of the presented Tables? This would enable one to determine the significance of any deviations from a baseline. This is particularly important here as some of the gains presented (e.g., in Table 2) are small enough that they could be attributed to noise.
5. **(Critical)** The use of bold in Table 1 is very confusing and seems to follow an unconventional use. Is it the case that only the best scores for each metric are in bold as it is traditionally done? If so, then why are there no entries in bold for the Task accuracy (where MCBM underperforms), and why are certain MCBM results bolded when, in fact, they are worse than competing baselines (e.g., “explanation” for CUB and MIMIC-II)? I think it is absolutely okay to use bolding for any purpose as long as it is made clear to the readers. In the absence of an explanation for it, however, I would say the common assumption is that bold fonts indicate the best-performing baseline (which does not appear to be the case here).
6. **(Critical)** In Section 1, it is claimed that information leakage may affect the ability to intervene on CBMs (a statement I agree with for vanilla CBMs but seems to not be the case for other sorts of CBM-like models as recent evidence suggests [1]). Is it the case that interventions in MCBMs lead to higher accuracies than in their CBM counterparts? Given the importance that interventions have for CBMs (and the way they serve as a verification of their interpretability across the literature), it would be extremely helpful to understand what they look like for MCBMs.
7. **(Major)** The appendices show that the hyperparameter $\texttt{msl}$ has a significant effect on the performance and interpretability of the resulting MCBM. How was this selected for the results shown in Section 5? How would one select this argument in practice?
8. **(Minor)** It is claimed that the method “does not introduce any computational overhead compared to a Sequential CBM with a single decision tree as label predictor”. This is true from an asymptotic complexity point of view, but it may not necessarily be true from a practical point of view (asymptotic analysis does not consider the average instance and ignores potentially large constants, which may have non-trivial effects in the "small n" limit). In practice, what is the observed overhead in training an MCBM vs a CBM as the number of samples or concepts varies?
9. **(Minor)** If my understanding is correct, from Algorithm 1 and Section 4.1’s description, the concept predictor’s outputs are used during test time in both the global and the specialized decision trees. If that is true, then Figure 2 is a nice addition but may be a bit misleading as it appears to indicate that some of the concepts don’t come from the concept predictor, but instead, they come from some oracle/ground truth source. Is my understanding of how this method operates correct? If so, then why are there no connections/edges from the concept predictor to the hard CBM part (LHS) of Figure 2?
10. **(Minor)** How are the 45 concepts for CUB chosen? I can see the list of selected concepts in the Appendix, but it is unclear why these were selected over the rest.
11. **(Minor)** Out of curiosity, in case this has already been tried, if one does leaf-node specialization based on the **concept** **logits** rather than the probabilities, do you get better performance on incomplete datasets? If so, then why would this not be a better path than using the probabilities? Logits can also be calibrated and interpreted as probabilities, and they may enable more leakage that can benefit the downstream task.

### Minor Suggestions and Typos

Whilst reading this work, I found the following potential minor issues/typos which may be helpful when preparing a new version of this manuscript:

1. **(Potential Typo)** In line 80, “… the purpose of this work is provide …” should probably be “… the purpose of this work is to provide…”
2. **(Potential Typo, nitpicking)** In line 135, should “categorical vector” be “binary vector” instead for the concept vector $c$?
3. **(Potential Typo)** In line 141, “e.g” should probably be “e.g.”
4. **(Potential Typo)** In line 214, the citation to Platt is accidentally all upper-cased.
5. **(Formatting)** When using the opening quotations (”) in Latex, I would suggest using `` rather than ". Otherwise, the left quotation symbol is reversed (see Section 5.1 for examples).

## References

- [1] Zarlenga et al. "Learning to Receive Help: Intervention-Aware Concept Embedding Models." NeurIPS (2023).

---

> ### Author Response · Authors · 2024-11-20
> **Author Response**
>
> We sincerely thank the reviewer for reading our work and providing comments. We plan to respond to all highlighted points.
>
> **Weakness 1**:  *Moreover, it is unclear how MCBMs compare to and how they could be used to analyze much more modern approaches (e.g., Post-hoc CBMs, CEMs, ProbCBMs, Energy-free CBMs, etc.)*.
>
> **Response**:
> We understand this concern. In our Related Work of section 2, we provided a short justification of why we believe these modern CBM approaches (some of which you also mention) either do not address information leakage at all or they partially resolve the issue. Since reviewer MJtJ had a similar concern, we provide below the same detailed justification for three of these works:
>
> * *“Concept Embedding Models. NeurIPS 2022”*: This paper introduces the idea of a “concept embedding”, which was later adopted by more CBM papers such as PCBM [2]. In our work, we use scalar-valued concepts instead of concept embeddings, following the original CBM paper*. While concept embeddings achieve excellent task performance because they lead to more expressive concept representations, the authors do not comment on information leakage, i.e. they do not provide a justification about whether these concept embeddings also capture unintended (“leaked”) information from the inputs to improve the task performance. Instead, they propose the Concept Alignment Score (CAS) to measure how much learnt concept embeddings can be trusted as faithful representations of their ground truth concept labels. Their intuition is that clustering samples based on a faithful concept embedding would result in coherent clusters. While they show that CEMs achieve high CAS scores, we argue that their method may not be sufficiently interpretable because:
>     - They achieve CAS scores of around 80% in certain datasets, such as CUB and CelebA, which may imply the presence of leakage.
>     - This approach does not indicate which subsets may suffer from the imperfect concept alignment, or how does this imperfection affect concept-based explanations. In contrast, our tree-based method allows for group-specific leakage examination in the form of decision paths, and gives the exact decision rules based on leakage.
>     - The information captured in a high-dimensional concept embedding is unintuitive compared to a scalar-valued concept, which directly represents the probability (confidence) of the concept predictor.
>
> * *“Post-hoc Concept Bottleneck Models. NeurIPS 2022”*. Similar to CEMs, this work uses concept embeddings but in the form of concept activation vectors (CAVs). Also, the authors do not address the issue of concept faithfulness or that of leakage, since they rely on multi-modal models to learn concepts that may be unavailable. While they effectively deal with the problem of missing concept annotations, their concept quality and faithfulness relies on the fidelity of their multimodal model.
>
> * *“Addressing leakage in concept bottleneck models. NeurIPS 2022”*. This work is closer to our method, in the sense that a) it uses scalar-valued concepts and b) specifically addresses leakage. However, as we mention in our related work, “Havasi et al. (2022b) tackle missing information with a side channel and an auto-regressive concept predictor, but these approaches struggle with interpretability and disentanglement of residual information (Zabounidis 2023)”. Specifically, all such works which use a residual layer or side-channel aim to let missing concept information pass directly from inputs to targets, keeping the ground truth concept representations not influenced by leakage. Yet, (Zabounidis 2023) highlight that the residual is not guaranteed to capture this intended missing information, and the two representations may be entangled. We argue that the lack of transparency in the residual channel does not make these methods convincing enough for this problem.
>
> Thus, we believe that MCBM-Seq is more effective as an analysis tool and provides some novel advantages compared to existing methods, such as the ability to perform group-specific leakage examination and to identify the exact decision rules based on leakage. Our work cannot be compared with these previous works in terms of **how they deal with leakage**, since they either do not address this problem, or they indirectly address it using vastly different approaches. Also leakage was not quantitatively defined in these works in order to be properly compared (in contrast to other traditional metrics such as task accuracy).
>
> Regarding the question of whether MCBM-Seq **could instead be used to analyze these modern approaches**, we believe that it is compatible with many other CBM methods because it does not pose any constraint on the architecture of the concept encoder (lines 536-538). It is promising to combine an Auto-Regressive Concept encoder from the work “Addressing leakage in concept bottleneck models. NeurIPS 2022” with our tree-based label predictor.

---

> ### Author Response · Authors · 2024-11-20
> **Author Response (Continue)**
>
> **Weakness 1, Questions 1 and 2**: *The performance difference between MCBM-Seq and joint CBMs, arguably a weak baseline for incompleteness compared to more recent approaches, seems to be large enough that one could construct a very convincing argument that any gains in reductions in concept leakage are not worth it in practice...*
>
> **Response**
>
> Your concern about the significant drop compared to joint CBMs is perfectly reasonable. However, on the other hand joint CBMs highly suffer from information leakage as shown in previous work [3,4]. We believe that having a high-performing CBM with very unreliable explanations due to leakage contradicts the reason CBMs were created in the first place, which is to provide concept-based explanations. Otherwise, it would make more sense to use a high-performing black-box model. Similar to reviewer kyiD, we will attempt to describe the results of Tables 1 and 2 in this response in an intuitive and detailed manner, and we hope our argument will be clarified at the end of this response.
>
> The purpose of Table 2 is to show that MCBM-Seq is comparable in task performance compared to existing CBMs, or lower in performance compared to Joint CBMs which however suffer from information leakage (refer to the work of [3]) and Black-Box neural networks which are inherently uninterpretable. The Table does not show the advantage of our method by itself, but shows how it performs compared to standard methods in order for our analysis to be complete.
>
> The important take-away from Table 2 when looking at the numbers is that the relationship of **task accuracies** in CBM modes is roughly the following: Hard, Independent < **MCBM-Seq** <= Sequential < **MCBM-Joint** (for small $\lambda_C$) < Joint (for small $\lambda_C$) < Black-Box. In contrast, the problem of leakage follows the opposite trend: Hard, Independent (No Leakage by definition [3,4]) > MCBM-Seq (has leakage, but this is inspectable and controllable by the decision maker) > Sequential CBM (has leakage according to [3,4], which is uninspectable, uncontrollable and affects all samples as stated in L460-462) > MCBM-Joint (has more leakage but this is again controllable) > Joint-CBM (typically it has the most leakage and this is uncontrollable, based on [3,4]). The two reverse trends show the trade-offs of CBMs.
>
> **Table 1 was constructed to highlight the advantages of MCBM-Seq. First**, the last column named “Leakage Inspection” emphasizes that MCBM-Seq is the only method that allows for Leakage Inspection, which is the novel property we introduce in this work. The existing completely soft sequential CBMs, regardless of their type of label predictor (Entropy-Net, Simple Decision Tree), typically have leakage, as shown in previous works [3, 4] but this leakage is neither easily inspectable nor controlled, which motivated our work. We also provide an intuition for this claim with a practical example in Appendix A.3, page 14. **Secondly**, the table reveals another advantage of MCBM-Seq when compared explicitly with a Purely Soft Sequential CBM using an Entropy-Net as a label predictor, which is that MCBM also achieves higher Explanation Accuracy and does not raise Fidelity issues, as explained in lines 416-426. This second advantage does not hold when compared to purely soft sequential CBMs using traditional decision trees, but the first main advantage of leakage inspection still remains.
>
> **The reasons why our leakage inspection metrics is useful** are those highlighted in section 5.3: **a)** we can analyze our model for specific decision paths (groups), and thus **b)** we can derive more meaningful group-specific explanations, since bi) the decision-maker has the flexibility to control the concept explanation based on the length of the decision path (lines 515-518) and bii) leakage will not impact all decision-making paths in a mixed CBM (lines 518-519).
>
> In conclusion, the argument of our work is the following: **If MCBM-Seq has a task accuracy between those of a Hard and a Sequential CBM (Table 2) but is superior in terms of explainability due to its leakage inspection property, which is shown in Table 1 and section 5.3 (pages 9 and 10), then we believe it is a useful training method for CBMs**.
>
> In terms of comparing with other modern CBMs, please refer to our previous response. We explain that these methods are only comparable in terms of task performance and not in leakage mitigation. Our method is indeed inferior in prediction accuracy compared to leaky CBMs such as Joint CBMs. However, we believe that leakage inspection and control is a crucial property that has not been thoroughly examined and is equivalently (or even more) important than predictive performance for interpretable models like CBMs.
>
> [3] Mahinpei, A., Clark, J., Lage, I., Doshi-Velez, F., & Pan, W. (2021). Promises and Pitfalls of Black-Box Concept Learning Models. ArXiv, abs/2106.13314.
>
> [4] Addressing leakage in concept bottleneck models. NeurIPS 2022

---

> ### Author Response · Authors · 2024-11-20
> **Author Response (Continue)**
>
> **Question 3**: *If the argument to be built is that MCBMs are better for post-hoc analysis than prediction, then I would say the paper should focus more on the analysis part and show how MCBMs can be used to lead to actionable changes/edits/insights that improve the underlying model somehow. … Therefore, do you have any empirical evidence of actionable changes derived from insights from MCBM that led to a model’s update improving its performance under a reasonable metric?*
>
> **Response**
>
> The argument is indeed that MCBMs are better for post-hoc analysis than prediction, which was clarified in more detail in our previous response. Similar to reviewer VU8H, in this response we will first justify more why we consider leakage inspection useful and then explain potential actionable changes proposed in section 5.3 in more detail.
>
> Let us first adapt a practical example of leakage in CBMs described in [1]. Assume we have a CBM performing animal classification. When the concept predictor recognizes an image as a dog, it may predict a slightly higher likelihood to the concept ‘tail’ compared to an image of a cat. Let us assume for example that the concept predictor gives a likelihood around 0.8 to the majority of dogs for the concept ‘tail’, and around 0.6 for the majority of cats. The ground truth concept ‘tail’ for both classes is 1 (meaning that both animals have a tail). Even though cats and dogs are indistinguishable based on this ground truth concept, the label predictor may still compare the two soft concept probabilities (0.6 and 0.8) and make an accurate classification. However, is this concept-based explanation useful? The label predictor relies on a difference in likelihoods that may (or may not) be interpretable to a human. For example, it might be the case that in most images of cats their tail is not clearly shown due to the orientation of the cat, whereas in most images of dogs the tail is clearly visible. This phenomenon is called “leakage” because unintended information from the input is captured in the concept likelihood (the orientation of cats and dogs). Thus, the concept predictor assigns a higher likelihood to dogs having this concept. However, this may be just an assumption. Regardless of whether this difference is intuitive or not, we believe **it is crucial to first find a systematic way to isolate instances** where the label predictor takes advantage of such differences, in order for us to then proceed to further analysis.
>
> **Thus, our work first provides a systematic way to identify those instances affected by leakage using our tree structure**. Specifically, we first train a tree using only the ground truth concepts, which we name the “global tree” (step 1 in Figure 1). Each leaf node in the global tree corresponds to a subset of examples that are indistinguishable based on their ground truth concepts. For example, if we observe the leftmost decision path in the toy MNIST example of Figure 3, there is a leaf node corresponding to group of 926 digits (278 “6”s, 33 “8”s and 615 “9”s), and all these digits have small length and small thickness. We see that this group was not extended with leakage after performing our algorithm, i.e. a sub-tree was not found (step 2 in Figure 1). **This means that the label predictor could not take advantage of any differences in the likelihood of concepts “length:small” and “thickness:small” to further split this group**. On the other hand, the third decision path from the left, which corresponds to the group of digits having small length and large thickness, was extended with two decision rules corresponding to differences in concept likelihoods (leakage): “length:small < 0.8” and “thickness:large < 0.8”.
>
> *(the answer to question 3 continues to the next response)*

---

> ### Author Response · Authors · 2024-11-20
> **Author Response (Continue)**
>
> *(we continue our response to question 3)*
>
> **Identifying such instances can then assist our decision-making when deriving concept-based explanations per decision path**. This corresponds to the analysis of section 5.3. For better clarity, we update this section by defining two scenarios:
>
> * If a group (leaf node) **cannot** be further split using leakage, i.e. a sub-tree is not found: This shows that the particular group is **not affected by information leakage, which is desirable**. In addition, if the task accuracy for this particular group is high, we may consider this an ideal classification example, because the available concept annotations are sufficient for an accurate distinction. The fact that we can identify and isolate such groups is highlighted as a key advantage of our work. **Unlike a purely soft CBM, leakage will not impact these groups, and thus the concept explanations for those groups are both leakage-free and accurate**. If, on the other hand, the task accuracy is not sufficient, we may flag this group to an expert to either annotate additional concepts or perform an intervention (future work).
>
> * If a group (leaf node) can be further split using leakage, i.e. a sub-tree is found: Then this group can be flagged for additional analysis. We describe a detailed case study for the Woodpecker example in page 10, paragraph: “Our tree-structure allows for meaningful group-specific explanations”. We urge the user to investigate if this difference in likelihoods (leakage) might be intuitive or not, similar to the cat-dog example we described in the beginning. Then the user has the following options: a) rely solely on the decision process of the global tree to derive a perfectly understandable, leak-free explanation, b) extend the decision process with the sub-tree of MCBM-Seq if this likelihood difference seems intuitive, c) use MCBM-Joint’s less intuitive probabilities for maximum accuracy, d) flag this group to an expert to either annotate additional concepts or perform an intervention (future work).
>
> In conclusion, identifying leakage is useful because it provides these analysis tools to a decision maker when deriving explanations, which are not available to a standard CBM. We currently develop future work providing interventions and concept discovery strategies specifically when leakage is observed. However, future leakage mitigation strategies first require a method that controls and inspects leakage for specific sub-groups, thus we believe this work may be used as a very useful analysis tool.

---

> > ### Comment · Reviewer_MYyQ · 2024-11-24
> > **Thank you for your rebuttal**
> >
> > Dear Authors,
> >
> > Thank you for taking all the time, effort, and patience to reply to some of my many questions and concerns. The time taken to do this is certainly appreciated. Below, I outline a few general comments after carefully reading your responses:
> >
> > - I entirely agree with the authors and understand the argument that competing methods may have more leakage and, therefore, better performance. However, my concern is that there must be some tolerance in performance drop for a model's utility to be worth it over competing baselines. For example, it is ok for a method to perform slightly worse than black box models if they can provide more things that one considers useful (e.g., interpretability, interventions, etc). However, even there, there is an implicit tolerance to how much one would be willing to sacrifice in performance before any benefits of the new approach are just simply not worth the drop. My argument here is that it seems that MCBM's drop in performance, even against very weak baselines to today's standards (e.g., Joint-CBMs are usually significantly outperformed by CEMs, Post-hoc CBMs, etc), would, in my opinion, be beyond that acceptable threshold for practical tasks.
> >
> > - Related to the point above, if MCBM  is sold as a predictive model, then I do not see a particularly strong case for why these more recent baselines are exempt from evaluation because they are more potentially "leaky". If that is the case, then the evidence will show that, and readers will also be able to place the proposed method with respect to more modern approaches that have gained significant momentum recently. Without an actual evaluation, it is hard to do this and to understand what MCBM brings to the table that other baselines do not.
> >
> > - Moreover, even if MCBM controls for leakage much better than competing approaches, at the end of the day, what the average practitioners are potentially most interested in is whether the explanation is aligned with what the model predicts at the end (something that is much easier to evaluate via interventions than by measuring leakage). Without any intervention evidence, it is hard to fully understand what I get in practice from reducing leakage. The theoretical/meta argument for why reducing leakage is potentially good for interventions is reasonable (although, as pointed out in my review, there is more recent evidence that challenges this notion). Yet, the argument for that could be clearer/stronger if empirical evidence is there to support the claim. I was hoping that this evidence could come as part of this rebuttal (as it does not require re-training, it is just evaluation). However, I understand that the rebuttal window is tight, and this may be left for future work.
> >
> > - I strongly suggest that my comments above on error bars and bolding be addressed in the next iteration of this manuscript, as they go against common good practices. Apologies if I missed a comment/change in your rebuttal that introduced these changes, though.
> >
> > - Perhaps more importantly, **I still believe that by presenting MCBM as both a predictive model and an analysis tool, this work is attempting to cover perhaps too much without constructing a particularly strong case for either of these two directions**. I am happy to be convinced otherwise by my fellow reviewers and ACs. However, from the paper and rebuttal itself, I am still not entirely convinced that either direction is strongly supported by evidence suggesting MCBM should be adopted over existing alternatives (this particularly goes for the predictive model side of the argument).
> >
> > Because of these reasons, **I am willing to increase my score slightly to a borderline reject but will not raise my score further as several of my critical concerns were not addressed during the rebuttal**.
> >
> > Once more, I thank the authors for their rebuttal and their paper and wish them the best of luck with this submission.

---

> ### Author Response · Authors · 2024-11-27
>
> Dear Reviewer MYyQ,
>
> We once again thank you for your comments. First, we would like to provide some further arguments on the problem of task accuracy drop, by referencing again to the state-of-the-art CBM models, and I hope this confusion will be resolved.
>
> Accuracy drop is empirically observed in practically **all CBMs** including the state-of-the-art models, when we compare joint to sequential training, and is often even more severe when we use independent training. For example, consider the following paper:
>
> "Post-Hoc CBMs" [1] : Since these CBMs first train the backbone independently, they can be considered an improved form of sequential CBM training with many advantages. The authors compared the performance of PCBM and a simple Joint CBM, and they state themselves in page 15 of the Appendix: **"CBMs achieve a slightly better performance than PCBMs, and the original backbone". This fact did not obstruct this paper from achieving a spotlight acceptance at ICLR 2023**, due to the numerous advantages of their architecture such as dealing with missing concept annotations.
>
> Carefully reviewing more such papers, the argument that accuracy drops between different modes of training can be claimed for most state-of-the-art CBMs, yet these works were accepted due to dealing with other CBM problems. In this paper, we try to follow a similar logic as well, presenting the advantage of leakage inspection while not observing significant drop between CBMs **with the same training mode**, e.g. MCBM-Seq with Sequential CBM. The task accuracy comparison of MCBM-Seq with Joint CBM is not fair. We could have put the comparison with joint training only in the Appendix similar to Post-hoc CBMs, if this is such an issue.
>
> Regarding the concerns we did not yet reply, specifically those regarding **interventions** and **lack of error-bars**, we absolutely agree and we were working on those experiments during the rebuttal period. However, due to the tight deadline of updating the pdf, we were unable to do so and we decided to **withdraw the paper** and include these on a future version of our paper.
>
> [1] Post-hoc Concept Bottleneck Models. ICLR 2023

---

### Official Review · Reviewer_kyiD · 2024-11-04

**Soundness:** 3
**Presentation:** 3
**Contribution:** 3
**Rating:** 5
**Confidence:** 4

**Summary:**

Concept-bottleneck Models (CBM) suffer from information leakage, where the model exploits information in the soft concept scores with unintended consequences towards interpretability.
The paper addresses the leakage issue by fitting constrained decision trees on top of the concept scores.
They quantify information leakage with a mutual information measure and propose a three-step procedure for modeling.
The paper argues that their technique yields better explanations even when the concept set is incomplete, and guides the developer to concept sets that needs expansion.

I found the idea neat and the presentation clear but found empirical validation underwhelming.

**Strengths:**

- The paper is well motivated and the presentation is clean. They argued their modeling choices well.
- Mixed-CBM idea is intuitive, and follows from their mutual information measure of (4).

**Weaknesses:**

**Empirical Validation**

The results in Table 1 and Table 2 indicate that MCBM-* (their method) has similar task accuracy to Sequential/Joint with vanilla decision trees.
MCBM-*, however, allows for leakage inspection as remarked in Table 1 or in Section 5.3.
I too see MCBM-*'s major contribution is with leakage inspection, but the paper makes only a sparing evaluation of the same.
I expect to see stronger evaluation of MCBM's utility for information leakage, and their implications in improving task accuracy or explanation quality.
Perhaps through human-studies or mining new concepts on one of the tasks such as CUB.

From Table 2, I do not see how MCBM-* is better. It has worser task accuracy than EntropyNet, but perhaps lower leakage? (which is not apparent).
Given that their method has advantages when the number of concepts is small, their evaluation too should bring out more readily.

Overall, I did not find the evaluation convincing on (a) the promise and implications of MCBM-*'s leakage inspection, (b) MCBM's merits over decision trees or any other baselines.

**Questions:**

1. Please explain the metrics at length. Concept accuracy, fidelity and explanation accuracy. I believe explanation accuracy is mentioned but never used (in which case it can be dropped).
2. From the argument in L460-462, decision trees with soft concept scores must have led to more leakage (or poor fidelity?), but that's not the case in Table 1. Please explain.
3. When the concept scores are mixed, I do not see why only the concepts on the path are softened. What happens when all the concepts are softened (when expanding the tree) or only the leaf concept is softened.
4. Please comment on the choice of hyperparam. Table 2 lists out numbers for various hparam $\lambda_C$, but how is it picked?
5. (Comment) A picture or description of Morpho-MNIST can be handy.

---

> ### Author Response · Authors · 2024-11-19
> **Author Response**
>
> We sincerely thank the reviewer for reading our work and providing comments. We will plan to respond to the highlighted weaknesses and questions.
>
> **Weakness 1**: *I expect to see stronger evaluation of MCBM's utility for information leakage, and their implications in improving task accuracy or explanation quality. Perhaps through human-studies or mining new concepts on one of the tasks such as CUB*.
>
> **Response**: We believe that section 5.3 focuses specifically on the implications of leakage on a) task accuracy and b) explanation quality, and c) it also provides a detailed case study on the CUB dataset.  More specifically:
>
> * *Impact on Task Accuracy*: As explained in paragraph “Our method enables inspecting Information Leakage per decision path”, the task accuracy increased in all three decision paths that were affected by leakage, for the reduced Morpho-MNIST example of Fig. 3. This increase shows exactly the impact of leakage, i.e. the task accuracy increases when a decision path of the Hard CBM is extended with additional decision splits relying on leakage. These numbers and paths are shown in Fig. 5 and Table 3. As an example, consider the decision path with number 14 in Figure 5. In the global tree of the Hard CBM, the decision path would end at the light gray node, which has 571 samples all having large length, large width and medium thickness. Due to majority voting, the hard CBM classifies all those digits as “8”s. As shown in Table 3, this leads to an accuracy of 44.91% for this path with number 14. When MCBM-Seq is then applied to investigate if this group is prone to leakage, the algorithm discovers that the group can be further split using the soft concept representation of “length:large”. Specifically, it is observed that the concept predictor is less than 90% confident that the majority of “6”s have a large length, while it is more than 90% confident that the majority of “8”s have this attribute. Thus, the tree-based label predictor takes advantage of this difference in predicted likelihoods, and is able to further split the group of 571 samples into two new groups using the decision rule “length::large <= 0.9”. With this new “leaky” split, the task accuracy improves to 57.70% for the path, as shown in Table 3.
>
> * *Impact on Explanation Quality*: As explained in paragraph “Our tree-structure allows for meaningful group-specific explanations”, identifying groups affected by leakage can assist our decision-making when deriving concept-based explanations per decision path. For better clarity, we update this paragraph by defining two scenarios:
>
>      - If a group (leaf node) **cannot** be further split using leakage, i.e. a sub-tree is not found: This shows that the particular group is not affected by information leakage, which is desirable. In addition, if the task accuracy for this particular group is high, we may consider this an ideal classification setting, because the available concept annotations are sufficient for an accurate distinction. The fact that we can identify and isolate such groups is highlighted as a key advantage of our work. **Unlike a purely soft CBM, leakage will not impact these groups, and thus the concept explanations for those groups are both leakage-free and accurate**. If, on the other hand, the task accuracy of the group is not sufficient, we may flag this group to an expert to either annotate additional concepts or perform an intervention (future work).
>
>    - If a group (leaf node) **can** be further split using leakage, i.e. a sub-tree is found: Then this group can be flagged for additional analysis. The user has the following options: a) rely solely on the decision process of the global tree to derive a perfectly understandable, leakage-free explanation. In the example of the decision path with number 14 described above, this means terminating the explanation at the light gray node and characterizing all 571 samples as “8”s. b) extend the decision process with the sub-tree of MCBM-Seq if this likelihood difference seems intuitive. In the same example, this means incorporating the path extension into our explanation, using the decision rule “length::large <= 0.9”. c) use MCBM-Joint’s less intuitive probabilities for maximum accuracy, d) flag this group to an expert to either annotate additional concepts or perform an intervention (future work).
>
> * *Case study on a real-world setting*: We also provided a detailed case study in Appendix A.10, page 24 on the CUB dataset, for distinguishing between the Red Bellied and the Red Headed Woodpeckers. Figure 15 shows the complete decision path (explanation) of the case study both before and after incorporating the path extension, and also highlights that the test accuracy again increases due to leakage for this path.
>
> Are there any specific additional experiments you would like to see? We would be happy to consider them.

---

> ### Author Response · Authors · 2024-11-20
> **Author Response (Continue)**
>
> **Question 1**: *Please explain the metrics at length. Concept accuracy, fidelity and explanation accuracy. I believe explanation accuracy is mentioned but never used (in which case it can be dropped)*.
>
> **Response**: As stated in lines 383-388, these are all metrics introduced in existing works. More specifically:
> * The *concept accuracy* introduced in [1] refers to the accuracy of the concept predictor when predicting all categorical concepts (refer to Step 1 of the method, Figure 1, page 2). Since each concept is predicted independently, the total concept accuracy reported is the aggregation of all predicted concepts. We can observe in Table 2, page 9, that the concept accuracy is the same for all CBM types where the concept predictor is trained independently. For Joint CBMs, we observe the trade-off between concept and task accuracy based on the parameter $\lambda_C$. This is explained in detail in the original CBM work of [1]. The Black-Box model does not have a concept accuracy since it does not use concept supervision (inputs are directly mapped to targets in a single neural network).
>
> * The *task accuracy* refers to the accuracy of the label predictor. For example, the task accuracy of the Hard CBM refers to the accuracy of the global decision tree (refer to Step 1 of the method, Figure 1, page 2), while the task accuracy of MCBM-Seq refers to the accuracy of the same tree but extended with all potential sub-trees (refer to Step 3 of the method, Figure 1, page 2).
>
> * The *explanation accuracy* introduced in [2] measures the task performance of a model when using its extracted explanation formulas instead of the model’s predictions, which would correspond to the task accuracy. In case of Entropy-Net, their method approximates the predictions of a neural network using simplified logic rules. Thus, the task accuracy is always greater or equal than the explanation accuracy when using Entropy-Net, as can be observed from Table 1, page 9 as well as the original results of the paper [2]. However, in decision trees, the explanation accuracy is the same as the task accuracy by default, since a decision path from the root to a leaf node serves as both the classifier and the explanation.
>
> * The *fidelity of an explanation* is also introduced in [2] and measures how well the extracted explanation matches the predictions obtained using the explainer. In practice, the authors calculate it as the accuracy score between the labels predicted from the neural network and the labels predicted from the logic rules that serve as explanations. In decision trees, we denote the fidelity score as 100% to indicate that no fidelity considerations occur, since a decision path from the root to a leaf node serves as both the classifier and the explanation.
>
> [1] Koh, P. W., Nguyen, T., Tang, Y. S., Mussmann, S., Pierson, E., Kim, B., & Liang, P. (2020, November). Concept bottleneck models. In International conference on machine learning (pp. 5338-5348). PMLR.
>
> [2] Barbiero, P., Ciravegna, G., Giannini, F., Lió, P., Gori, M., & Melacci, S. (2022). Entropy-Based Logic Explanations of Neural Networks. Proceedings of the AAAI Conference on Artificial Intelligence, 36(6), 6046-6054.

---

> ### Author Response · Authors · 2024-11-20
> **Author Response (Continue)**
>
> **Weakness 2 and Question 2**: *From Table 2, I do not see how MCBM is better. It has worser task accuracy than EntropyNet, but perhaps lower leakage? (which is not apparent). Given that their method has advantages when the number of concepts is small, their evaluation too should bring out more readily. From the argument in L460-462, decision trees with soft concept scores must have led to more leakage (or poor fidelity?), but that's not the case in Table 1. Please explain*.
>
> **Response**: We apologize if the interpretation of Tables 1 and 2 is not immediately clear. We will attempt to describe them in this response in an intuitive and detailed manner.
>
> The purpose of Table 2 is to show that MCBM-Seq is comparable in task performance compared to existing CBMs, or lower in performance compared to Joint CBMs which however suffer from information leakage (refer to the work of [3]) and Black-Box neural networks which are inherently uninterpretable. The Table does not show the advantage of our method by itself, but shows how it performs compared to standard methods in order for our analysis to be complete.
>
> The important take-away from Table 2 when looking at the numbers is that the relationship of *task accuracies* in CBM modes is roughly the following: Hard, Independent < **MCBM-Seq** <= Sequential < **MCBM-Joint** (for small $\lambda_C$) < Joint (for small $\lambda_C$) < Black-Box. In contrast, in terms of leakage, they follow the opposite trend: Hard, Independent (No Leakage by definition [3,4]) > **MCBM-Seq** (has leakage, but this is inspectable and controllable by the decision maker) > Sequential CBM (has leakage according to [3,4], which is uninspectable, uncontrollable and affects all samples as stated in L460-462) > **MCBM-Joint** (has more leakage but this is again controllable) > Joint-CBM (has a lot of uncontrollable leakage, based on [3,4]). The two reverse trends show the trade-offs of CBMs.
>
> **Table 1 was constructed to highlight the advantages of MCBM-Seq**. **First**, the column named “Leakage Inspection” emphasizes that MCBM-Seq is the only method that allows for Leakage Inspection, which is the novel property we aim to introduce in this work. The existing completely soft sequential CBMs, regardless of their type of label predictor (Entropy-Net, Simple Decision Tree), typically have leakage, as indicated in previous works [3, 4] but this leakage is neither easily inspectable nor controlled, which motivated our work. We also provide an intuition for this claim with a practical example in Appendix A.3, page 14. Hard and Independent CBMs are not included in this table because they do not have leakage, so leakage inspection is not applicable. **Secondly**, the table reveals another advantage of MCBM-Seq when compared explicitly with a Purely Soft Sequential CBM using an Entropy-Net as a label predictor, which is that MCBM-Seq also achieves higher Explanation Accuracy and does not raise Fidelity issues, as explained in lines 416-426. This second advantage does not hold when compared to purely soft sequential CBMs using traditional decision trees, but the first main advantage of leakage inspection still remains.
>
> **The reasons why our leakage inspection is useful** are those highlighted in section 5.3: **a)** we can analyze our model for specific decision paths (groups), and thus **b)** we can derive more meaningful group-specific explanations, since bi) the decision-maker has the flexibility to control the concept explanation based on the length of the decision path (lines 515-518) and bii) leakage will not impact all decision-making paths in a mixed CBM (lines 518-519).
>
> Based on the above clarifications and going back to Question 2, the answer is that decision trees with purely soft concept scores do not raise fidelity issues but they also do not allow for Leakage Inspection, unlike our new MCBM-Seq method. We again encourage the reviewer to refer to Appendix A.3, page 14 which shows a decision tree with purely soft scores and how its inefficiency motivated us to develop MCBM-Seq.
>
> In conclusion, **the argument of our work is the following: If MCBM-Seq has a task accuracy between those of a Hard and a Sequential CBM (Table 2) but is superior in terms of explainability due to its leakage inspection property, which is shown in Table 1 and section 5.3 (pages 9 and 10), then we believe it is a useful training method for CBMs**. We hope this clarifies the use of our Tables.
>
> [3] Mahinpei, A., Clark, J., Lage, I., Doshi-Velez, F., & Pan, W. (2021). Promises and Pitfalls of Black-Box Concept Learning Models. ArXiv, abs/2106.13314.
>
> [4] Addressing leakage in concept bottleneck models. NeurIPS 2022

---

> ### Author Response · Authors · 2024-11-20
> **Author Response (Continue)**
>
> **Question 3**: *When the concept scores are mixed, I do not see why only the concepts on the path are softened. What happens when all the concepts are softened (when expanding the tree) or only the leaf concept is softened*.
>
> **Response**: This is a very important detail. We specifically refer to this in lines 258-268. If all concepts are softened, the mutual information for the leaf node: $I(y;\hat{c}_k|c_s)$ would not be satisfied, thus our Definition 3.1 of Leakage in line 176 would not hold since Equation (8) in the Appendix would be incorrect. Refer to Appendix A.2, page 14 for further details. Intuitively, the reason is that we treat Information Leakage as “The amount of unintended information that is used to predict label $y$ with soft concepts $\hat{c}$ that is not present in hard representation $c$. Thus, we need to make sure first that all samples of a group have the hard concept $c$, in order to then investigate if the soft representation of this concept provides **extra**, leaky information. **We cannot investigate the impact of a soft concept in a sample if the sample does not possess the hard concept in the first place.**
>
> The concepts appearing in the decision paths are guaranteed to be shared by all samples in the path. Taking the leftmost path in Figure 3, page 6 as an example, the 926 digits ending up in the green leaf node all have small length and small thickness, if we follow their decision path from the root. We then search if the soft representation of any of those two concepts leads to leakage, which does not happen in the particular path because there no nodes with light gray color were found. Concepts not appearing in the path may not be shared by all digits in the group.
>
> **Question 4**: *Please comment on the choice of hyperparam. Table 2 lists out numbers for various hparam $\lambda_C$, but how is it picked?*
>
> **Response**: According to the original CBM work [1], the parameter $\lambda_C$ controls the trade-off between task and concept accuracy in Eq. (3) line 165, which is also evidenced in our results of Table 2 (smaller values of this parameter increase the task accuracy and reduce the concept accuracy, while the opposite holds for large values). For computational reasons, we tested one very small value $\lambda_C = 0.1$, one very large value $\lambda_C = 100$ and one in the middle, to show how the metrics change in the full range of values for this parameter.
>
> [1] Koh, P. W., Nguyen, T., Tang, Y. S., Mussmann, S., Pierson, E., Kim, B., & Liang, P. (2020, November). Concept bottleneck models. In International conference on machine learning (pp. 5338-5348). PMLR.

---

### Note · Authors · 2024-11-27

**Comment:**

Dear Reviewers,

We would like to express our sincere gratitude for the time and effort you have dedicated to reviewing our submission. After careful consideration, we have decided to withdraw our paper from the review process. The reason is that we aim to make notable changes to the experimental section of the paper, and thus we expect the new pdf to be relatively different. Such changes include a) an intervention analysis and b) a more thorough explanation of the usefulness of leakage inspection, related to the answers we provided on your comments.

Best regards, Submission10191 Authors

**Withdrawal Confirmation:**

I have read and agree with the venue's withdrawal policy on behalf of myself and my co-authors.